# AGMMU: A Comprehensive Agricultural Multimodal Understanding Benchmark

**Aruna Gauba**[1,2,5*]  **Irene Pi**[1,3,5*]  **Yunze Man**[1,4,5†]  **Ziqi Pang**[1,4,5†]

**Vikram S. Adve**[1,4,5]  **Yu-Xiong Wang**[1,4,5]

[1]University of Illinois Urbana-Champaign  [2]Rice University  [3]Carnegie Mellon University

[4]AIFARMS  [5]Center for Digital Agriculture at UIUC

[*†] Equal Contribution  [†] Project Lead

## Abstract

We present **AGMMU**, a challenging *real-world* benchmark for evaluating and advancing vision-language models (VLMs) in the knowledge-intensive domain of agriculture. Unlike prior datasets that rely on crowdsourced prompts, AGMMU is distilled from *116,231 authentic dialogues* between everyday growers and *USDA-authorized Cooperative Extension experts*. Through a three-stage pipeline: automated knowledge extraction, QA generation, and human verification, we construct (i) **AGMMU**, an *evaluation set* of 746 multiple-choice questions (MCQs) and 746 open-ended questions (OEQs), and (ii) **AGBASE**, a *development corpus* of 57,079 multimodal facts covering five high-stakes agricultural topics: insect identification, species identification, disease categorization, symptom description, and management instruction. AGMMU has three key advantages:

- **Authentic & Expert-Verified**: All facts, images, and answers originate from real farmer and gardener inquiries answered by credentialed specialists, ensuring high-fidelity agricultural knowledge.
- **Complete Development Suite**: AGMMU uniquely couples a dual-format evaluation benchmark (MCQ *and* OEQ) with AGBASE, a large-scale training set, enabling both rigorous assessment and targeted improvement of VLMs.
- **Knowledge-intensive Challenge**: Our tasks demand the synergy of nuanced visual perception and domain expertise, exposing fundamental limitations of current general-purpose models and charting a path toward robust, application-ready agricultural AI.

Benchmarking 12 leading VLMs reveals pronounced gaps in fine-grained perception and factual grounding. Open-sourced models trail after proprietary ones by a wide margin. Simple fine-tuning on AGBASE boosts open-sourced model performance on challenging OEQs for up to 11.6% on average, narrowing this gap and also motivating future research to propose better strategies in knowledge extraction and distillation from AGBASE. We hope AGMMU stimulates research on domain-specific knowledge integration and trustworthy decision support in agriculture AI development.

**Code**: https://github.com/AgMMU/AgMMU
**Data**: https://huggingface.co/datasets/AgMMU/AgMMU_v1

39th Conference on Neural Information Processing Systems (NeurIPS 2025) Track on Datasets and Benchmarks.

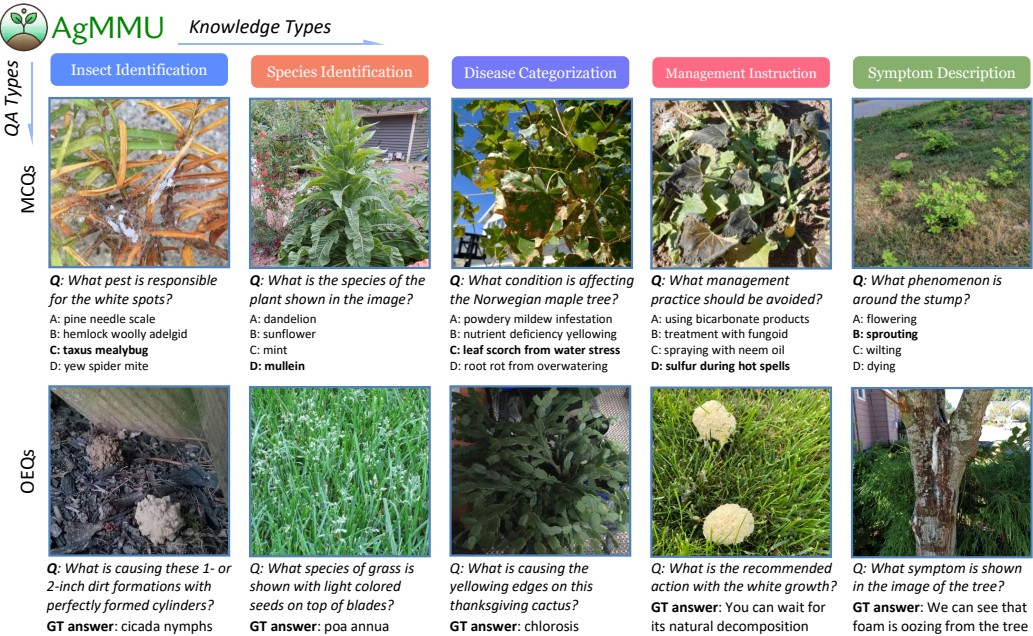

Figure 1: **AGMMU** is a multimodal agricultural dataset that challenges vision-language models (VLMs) to observe the details of images and provide factually precise answers. Derived from real-world conversations between users and authorized experts by USDA-funded Cooperative Extension, AGMMU covers five major agricultural knowledge types (demonstrated in five columns of the figure). AGMMU features 746 *multiple-choice questions* (MCQs) like conventional vision-language benchmarks [54] and the same number of *open-ended questions* (OEQs) like SimpleQA [51], all validated by human annotators. We also curate an agricultural knowledge base with 57,079 pieces of information for foundation model fine-tuning, extracted from experts' answers. AGMMU can benefit both knowledge-intensive VLMs and the social good of agriculture.

## 1 Introduction

Recent progress in large language models (LLMs) and vision language models (VLMs) has demonstrated remarkable capabilities in general knowledge understanding, as exemplified by the outstanding performance on a variety of multimodal understanding benchmarks [21, 30, 35, 37, 51, 54]. However, existing benchmarks often emphasize general-domain tasks and may not fully capture the limitations of current models in more *specialized*, *knowledge-intensive* domains.

Agriculture, a cornerstone application of scientific biological knowledge, presents a particularly challenging domain and has been extensively investigated in the vision and machine learning community since the early years [14, 16, 17, 19, 29, 39]. Unlike common tasks in the general domain, agricultural problems often require both precise visual analysis (*e.g.*, identifying specific diseases, pests, or plant conditions) and extensive domain knowledge (*e.g.*, treatment protocols, growing conditions, and management practices). Visual data is exceptionally crucial for agriculture, because many serious problems (*e.g.*, pests, disease, nutrient stress, water stress) can only be recognized visually from nuanced appearances, such as the color and shape of leaves. The stakes are also high in this scenario: accurate and timely agricultural diagnostics can mean the difference between crop survival and failure, directly impacting food security and farmer livelihoods. Despite the critical nature of this domain, *we lack comprehensive benchmarks to evaluate and improve the agricultural understanding abilities of VLMs*, especially the *concerns from real users*.

The challenge of creating such benchmarks is multifaceted. First, biological and agricultural tasks, where computer vision is often applied, are notably labor-intensive for data collection. They require expert-labeled data specific to the target task. Meanwhile, agricultural expertise is scarce and specialized, making it difficult to curate high-quality evaluation data [9]. Second, real-world agricultural problems are inherently multimodal requiring both visual and textual understanding, combining the visual observation with background information. Third, no clear protocol defines a representative distribution of realistic agricultural questions. These challenges have left a significant gap in our understanding of how well current AI systems can handle real-world agricultural problems [2, 6, 7, 10, 12, 40].

In this work, we introduce the **AGMMU** (Agricultural Multimodal Understanding Benchmark), the *first real-world derived agricultural benchmark* designed to evaluate the capabilities of multimodal foundation models. Our benchmark leverages 116,231 real-world conversations between 2013-2024 hosted by Cooperative Extension [15], which offers one-to-one conversations between *real-world users* capturing images from their own devices and *professional agricultural experts funded by USDA to provide answers*. This real-world data source ensures that our benchmark captures authentic challenges which farmers and gardeners face every day, including complex visual symptoms, varied environmental conditions, and nuanced management decisions.

For data curation, we design an automatic pipeline to extract the agricultural knowledge (as in Fig. 2) from long-form expert answers and then hire human annotators to verify the quality of the final question-answer pairs. Based on our observation of the real-world questions, the final dataset covers five major types of agricultural questions and knowledge: insect identification, species identification, disease categorization, management instruction, and symptom description, as in Fig. 1.

Evaluations of a diverse collection of VLMs have revealed that existing foundation models struggle with knowledge-intensive agricultural scenarios, suffering mainly from insufficient knowledge. (refer to Sec. 4 for more details). Moreover, with our preparation of a large multimodal knowledge base, the open-sourced model can be notably improved with plain fine-tuning, even exceeding strong closed-source models in almost all question subdomains. Such observations signify the value of our agricultural benchmark and also encourage exploration of better strategies than simple fine-tuning.

Our contributions include:

1. **AGMMU**: A carefully curated evaluation set of 746 multiple-choice questions (MCQs) and the same number of open-ended questions (OEQs, where no candidate choices are provided) extracted from real-world conversations between users and USDA-funded experts and verified by our human annotators. Each extracted QA is designed to test both visual understanding and knowledge application in agricultural contexts.

2. **AGBASE**: A comprehensive agricultural multimodal knowledge dataset of 57,387 pieces of facts extracted from the long-form experts answers, coming from non-overlapping conversations from the evaluation set. This is suitable for improving VLM performance on agricultural tasks.

3. A systematic evaluation of the leading VLMs reveals their limitations in handling knowledge-intensive agricultural queries, along with an error analysis. In the future, **AGMMU** will provide a complete training and evaluation suite for investigating such problems.

## 2 Related Work

**Multimodal Foundation Models and Benchmarks.** The evolution of multimodal foundation models has been accompanied by increasingly sophisticated evaluation benchmarks [23] concentrating on a evolving set of problems significant for AI applications. Early benchmarks like VQAv2 [18] and GQA [21] focus on basic visual question-answering capabilities. ScienceQA [35] specifically targeted scientific reasoning with visual components, while Eyes-Wide-Shut [48] evaluates the model's ability to avoid hallucination. Recent benchmarks have emphasized real-world applications and complex reasoning. RealWorldQA [53] tests the ability of the models to handle practical, everyday visual queries. The MMMU [54] benchmark represents a comprehensive effort to evaluate multimodal understanding across multiple domains and task types. On top of these benchmarks, AGMMU presents the first real-world oriented vision-language benchmark targeting agriculture, a domain that faces severe data scarcity and lacks domain experts for large-scale dataset curation.

**AI in Agriculture.** AI has been extensively applied in the biological and agricultural domains [3, 33, 49, 55, 56, 57, 58], with significant datasets and benchmarks driving progress in species identification and disease classification [13, 27, 28, 38, 52], answering crop science questions [36, 55], and knowledge retrieval [5]. In terms of the multimodal perspective of agriculture, the iNaturalist dataset [19] marked a milestone by providing millions of species observations from citizen scientists, enabling the creation of a dataset and a deep neural network model (InsectNet) for the identification of pests [8, 14]. This was followed by more specialized collections like BioScan-1M/5M [16], which focused on microscopic biological images, and TreeOfLife-10M [39], which expands the data in a comprehensive phylogenetic framework. These datasets have facilitated numerous advances in automated species identification, disease detection, and biological image analysis. With the advance of vision-language models (VLMs) in recent years and farmers' need for direct

| Datasets | Type | Multimodal | Training | Expert | Factuality |
|---|---|---|---|---|---|
| SimpleQA [51] | OEQ | ✗ | ✗ | ✗ | ✓ |
| ScienceQA [35] | MCQ | ✓ | ✓ | ✓ | ✗ |
| MMMU [54] | MCQ | ✓ | ✗ | ✗ | ✗ |
| Our AgMMU | MCQ+OEQ | ✓ | ✓ | ✓ | ✓ |

| Datasets | Type | Multimodal | Real-world | Factuality |
|---|---|---|---|---|
| iNat21 [19] | CLS | ✗ | ✗ | - |
| CROP [55] | MCQ | ✗ | ✗ | ✗ |
| CDDM [28] | OEQ | ✓ | ✗ | ✓ |
| Our AgMMU | MCQ+OEQ | ✓ | ✓ | ✓ |

(a) Comparison with general benchmarks.    (b) Comparison with agricultural benchmarks.

Table 1: Objective of AgMMU. (a) AgMMU provides a *multimodal* benchmark in *expert* domains with a *training* set. With open-ended questions (*OEQs*), AgMMU emphasizes the *factual* accuracy of models, where a model has to recall precise facts without relying on options. (b) Besides, Ag-MMU is unique in leveraging questions from *real-world* users instead of recruited annotators. These properties make AgMMU a unique benchmark for advancing the VLMs for agriculture.

analysis of images, multimodal agricultural benchmarks emerge [28, 55] and evaluate the capabilities of VLMs to address agricultural problems. However, a major limitation of these benchmarks is that their images and questions are curated from agricultural experts instead of real-world users with agricultural difficulties, which cannot capture the distribution and complexities of real-world plant growing. From this perspective, our AGMMU fills the gap by building from the conversations between real-world users and experts.

## 3  The AGMMU Benchmark

### 3.1  Overview

**Objective.**    We build AGMMU, short for "Agricultural Multimodal Understanding," targeting the major challenges for VLMs to serve users in agricultural domains: *how can VLMs provide precise factual knowledge for real-world questions*? This objective leads to two critical design choices for our data curation. (1) **Real-world distribution.** The questions, images, and answers are derived from conversations between real users and experts, rather than being curated from the web or textbooks by annotators. (2) **Factual questions.** To evaluate the factual precision of models, we follow SimpleQA's [51] style by requiring the VLMs to answer open-ended questions (OEQs) in short phrases, directly mentioning the key knowledge helpful for the users. In addition to open-ended questions, we provide multiple-choice questions (MCQs) to align with most multimodal benchmarks like MMMU [54]. Our OEQs and MCQs combination provides a comprehensive evaluation.

**Data source.**    We carefully select the data source to accomplish the above objectives. AGMMU is curated from 116,231 real-world conversations between 2013-2024 hosted by Cooperative Extension [15]. Such a data source offers the following critical advantages: (1) **Real-world users**: the questions and images are posted by real users who have agricultural questions; (2) **Authorized experts**: USDA funds the cooperative extension to authorize a group of professional agricultural experts from universities to answer these questions. Only certified experts are allowed to provide answers, which ensures the quality of data; (3) **Eligibility for research**: With the Cooperative Extension funded by USDA and being a non-profitable organization, the data can be open-sourced for research purposes under the CC-BY-SA4.0 License. Pre-processing is executed to remove personal and confidential information from our release dataset.

**AgMMU Overview.**    We design pipelines involving both automatic processing and human verification to turn raw conversations into high-quality QA pairs for agriculture knowledge evaluation. After filtering and processing, our AGMMU comprises of: (1) an evaluation set with 746 MCQ and the same number of OEQ questions covering the major agricultural knowledge types (as in Fig. 1); (2) a larger scale development set with 57,079 QA pairs formatted as open-ended QA, which is disjoint with the evaluation set and aimed to support the fine-tuning of VLMs for agricultural knowledge. We compare AGMMU with related AI and agricultural benchmarks in Table 1, marking its uniqueness. Compared with the previous agricultural benchmarks, which primarily focus on a limited range of species or topics, *e.g.*, CROP [55] mainly covers rice and corn, and CDDM [28] covers 16 crops and 60 diseases, while our AgMMU covers 10164 species, 5071 pests, and 4545 diseases.

### 3.2  Dataset Curation

**Curation Pipeline.**    Each real-world conversation is a multi-turn QA between the user and expert, containing one or several pieces of factual agricultural knowledge. We design a four-step pipeline to convert such a conversation into one QA pair suitable for the VLM evaluation as in Fig. 2. **(1) Categorization.** We first tag user-expert QAs with an agriculture domain and extract the primary plant organism of interest. **(2) Knowledge Extraction.** Based on the domain tag, we further decompose each expert-user conversation into its most important pieces of agricultural knowledge, where our prompts vary for different domains tagged in the previous step. **(3) QA Generation.** This step is for

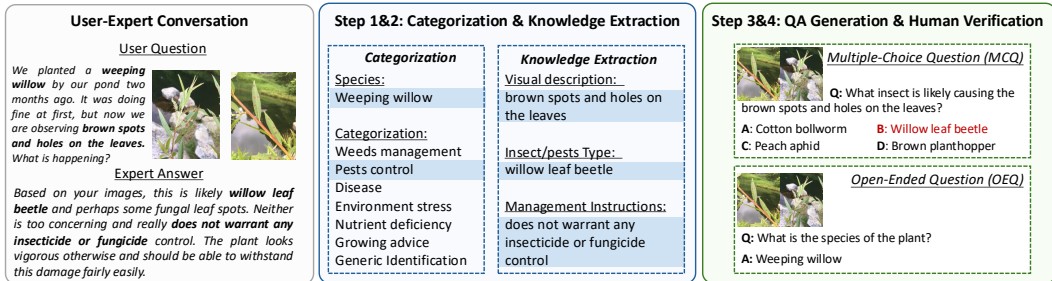

Figure 2: Starting from raw user-expert conversations, we design a four-step data curation pipeline with carefully designed prompts and human verification. (1) We employ LLaMA-70B to categorize the conversation and filter out the samples that do not fall under our selected *agriculture sub-domains*. (2) A larger LLaMA-405B model extracts key agricultural knowledge from the long-form user-expert conversation. (3) These facts either go to the evaluation set or the development set. For evaluation questions, we utilize GPT4o to format the original QA and agricultural knowledge into multiple-choice questions (MCQs) and open-ended questions (OEQs). (4) Finally, human annotators verify the quality of the questions and only keep the qualified ones in the evaluation set.

the evaluation set. We convert extracted agricultural facts into multiple-choice questions (MCQs) or open-ended questions (OEQs) for VLM evaluation. **(4) Human Verification.** We hire human annotators to filter out problematic QAs in the evaluation set.

**Step 1: Categorization.** Real-world agricultural questions come in a great variety with an open-ended nature. To extract clear agricultural knowledge, we must first account for the inherent complexity of these questions. Therefore, we first tag each question with an *agriculture sub-domain*. The seven categories we decide on are ["*disease advice*", "*weed management*", "*pests control*", "*growing advice*", "*environmental stress*", "*nutrient deficiency*", "*generic identification*", and *other*'], following the main categories designed by gardening and agriculture sites [31, 32, 43, 45, 46]. In this step, we filter out conversations in the "growing advice" and "other" sub-domains, as these are typically not image-related questions. Notably, the type of knowledge in user questions and expert answers often depends significantly on the question's sub-domain, which we use to guide the next step, where prompts are selected according to the specific agricultural sub-domain.

**Step 2: Knowledge Extraction.** We extract agricultural knowledge for training and evaluating VLMs without introducing redundant languages from the original conversations, as in Fig. 2. Across all questions, the user-expert conversation generally comprises a combination of identification (*e.g.*, disease, or plant species) and management instructions. For example, in Fig. 2, the user provides species identification information and symptom description, while the expert recognizes the diseases and offers management suggestions. To systematically extract knowledge, we organize the agricultural knowledge into categories of "*species identification*", "*disease identification*", "*symptom/visual description*", "*management instructions*", and "*insect/pest identification*."

To increase the reliability of information extraction, we leverage the agriculture sub-domain predicted in the previous step and design different prompt templates for each sub-domain. Our prompt leverages the following techniques: (1) Manually annotated in-context examples, which significantly improve knowledge extraction quality and avoiding hallucination. (2) Explicit fact verification, which determines whether the expert has provided relevant facts for the target category. We discover that LLMs [1, 11] can reliably complete the information extraction step. More details of our prompts and observations are available in the supplementary materials.

**Step 3: QA Generation.** For rigorous evaluation, AGMMU adopts multiple-choice questions (MCQs) and open-ended questions (OEQs), combining the advantages of MMMU [54] and SimpleQA [51], respectively. For the knowledge extracted from the previous step, a small and balanced subset is selected for evaluation, while the rest goes directly to **AGBASE** for model development and fine-tuning. We use GPT-4o [1] to format agricultural facts into QA pairs.

The following criteria guide the selection of the evaluation set via our close observation of the conversations. (1) Filtering is done by removing the conversations *without* management information or issue identification, as we empirically find that the absence of such information indicates expert uncertainty. We also filter out questions with uncertain terms, such as "unclear" in the extracted information. (2) We remove questions with management instruction terms that direct users to an external source rather than giving them an actual answer, typically containing the words "lab," "ar-

borist," etc. (3) To balance between different *agriculture sub-domains* (step 1) and *knowledge types* (step 2), we prioritize an even distribution across the types of questions, while maximizing the diversity of categories, as shown in Fig. 3.

Then we employ GPT4o [1] to convert these facts into MCQs and OEQs, without losing the original intention of the user's question. Depending on the *knowledge types* of the fact, we give the LLM a unique prompt to construct a corresponding question. We prompt it to generate three wrong answers and use the extracted fact as the ground truth to form one correct answer.

**Step 4: Human verification**  To guarantee the quality of the evaluation questions, we hire human annotators to manually verify the QA pairs from the following perspectives: (1) **Faithfulness**: whether the QA pair is faithful to the original conversation; (2) **Certainty**: whether the expert is certain about the answer; (3) **Quality**: are the images of good quality and correspond to the symptoms described by the user and the expert; (4) **MCQ Feasibility**: are all options other than the ground truth wrong. We strictly keep the data that pass all the evaluation criteria, resulting in 746 highest-quality MCQ and OEQ questions in **AGMMU** from 1742 samples.

**Ethical Compliance.**  We use automatic procedures assisted with human verification to erase the information that could leak the name, gender, address, or any other personal information for both users and experts. We will only distribute the processed AGMMU instead of the source data from the original conversation to protect the privacy of the users and experts. To the best of our verification, we have not found any human faces in the benchmark images.

## 3.3   Additional Properties of AGMMU

**Distribution and Coverage.**  In Fig. 3, we show the distribution of agricultural subdomains and knowledge types in AG-MMU. We have two key observations: (1) The original conversations we obtained are severely skewed in subdomains: Subdomains like nutrient deficiency and generic identification domains have significantly fewer samples than dominating domains such as disease advice or growing advice. This imbalance is inevitably carried over to knowledge types after the knowledge extraction stage. (2) However, in AG-MMU, we demonstrate a much more balanced distribution with our dataset curation pipeline. This is exceptionally important for our evaluation set, where our aim is to propose an all-encompassing benchmark that examines different aspects of models without inductive bias.

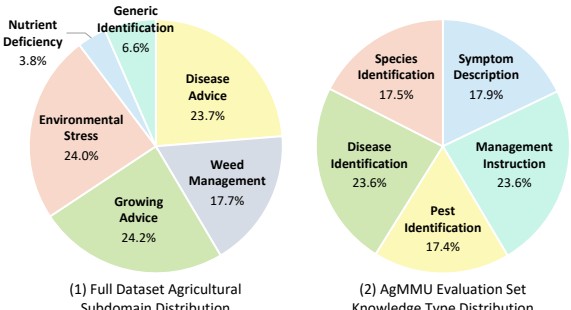

(1) Full Dataset Agricultural Subdomain Distribution

(2) AgMMU Evaluation Set Knowledge Type Distribution

Figure 3: **Statistics of AGMMU**. (1) The agricultural subdomain distribution of our raw dataset, after the categorization step, as explained in Figure 2. (2) AGMMU, after the knowledge extraction and evaluation curation steps, serves as a balanced subset of raw dataset with proportional representation across knowledge types.

**Realistic Images.**  Our AGMMU also features significant challenges in the quality and number of images. The images in our benchmark are generally uploaded by users in various qualities, resolutions, and aspect ratios. This differs from the photography-level images in datasets like iNaturalist [19] or the textbook or web-document figures like manually curated CROP [55] and CDDM [28]. As shown in Fig. 1, AGMMU realistically reflect the multimodal everyday scenarios encountered by gardeners and farmers. Our benchmark also exhibits the necessity of *multi-image understanding* challenge, since the user uploaded several images and the experts rely on a joint reasoning of them to make the final conclusion.

## 4   Experiments

We perform comprehensive evaluations on diverse vision-language models (VLMs), including both proprietary (closed-source) and open-sourced models. Our evaluation primarily focuses on zero-shot performance, utilizing either publicly available APIs or author-provided checkpoints to reflect the inherent capabilities of each model without task-specific training. Furthermore, we present fine-tuning results on LLaVA-1.5 to demonstrate the importance of our development set **AGBASE** for optimizing performance on agriculture tasks, and highlight some unique observations. All experiments are conducted using NVIDIA A6000 GPUs, and further details regarding model configurations, API choices, and implementation details are available in the supplementary material.

| | AgMMU-OEQs ↑ | | | | | | AgMMU-MCQs ↑ | | | | | |
|---|---|---|---|---|---|---|---|---|---|---|---|---|
| | Disease | Insect/Pest | Species | Management | Symptom | Average | Disease | Insect/Pest | Species | Management | Symptom | Average |
| *Proprietary Models* | | | | | | | | | | | | |
| GPT-4o | 43.8 | 49.6 | 58.2 | 24.2 | 25.7 | 40.3 | 80.7 | 91.6 | 88.9 | 90.8 | 81.5 | 86.7 |
| GPT-o4-mini [1] | 47.7 | 48.8 | 69.6 | 27.2 | 20.1 | 42.7 | 77.9 | 85.4 | 90.3 | 93.8 | 84.3 | 86.5 |
| Gemini 2.5 Pro | 50.0 | 61.1 | 77.1 | 29.6 | 20.5 | 47.7 | 84.9 | 88.2 | 93.1 | 92.6 | 85.5 | 88.9 |
| Gemini 1.5 Pro [34] | 37.6 | 51.7 | 69.9 | 40.1 | 23.3 | 44.5 | 76.2 | 81.1 | 82.8 | 88.1 | 76.9 | 82.4 |
| Claude 3.5 | 44.4 | 37.6 | 48.4 | 17.0 | 8.9 | 31.3 | 71.4 | 73.6 | 77.2 | 90.2 | 71.5 | 76.78 |
| Claude 3 Haiku [41] | 25.1 | 20.6 | 32.2 | 20.1 | 9.5 | 21.5 | 62.1 | 71.2 | 52.8 | 81.5 | 52.0 | 63.8 |
| *SOTA Open-sourced Models* | | | | | | | | | | | | |
| LLaVA-1.5-7B [25] | 4.9 | 22.3 | 30.6 | 9.2 | 16.5 | 16.7 | 61.9 | 64.6 | 67.6 | 77.3 | 71.5 | 69.0 |
| LLaVA-NeXT-8B [26] | 6.4 | 27.2 | 31.1 | 13.6 | 13.7 | 18.4 | 61.9 | 70.1 | 57.9 | 78.5 | 67.6 | 67.5 |
| LLaVA-OneVision-8B [22] | 9.1 | 24.1 | 34.1 | 13.1 | 15.3 | 12.3 | 65.7 | 72.9 | 71.2 | 85.9 | 78.8 | 75.4 |
| LLaVA-1.5-13B [25] | 8.2 | 22.1 | 31.8 | 13.9 | 10.9 | 17.4 | 64.7 | 67.4 | 65.5 | 80.4 | 73.7 | 70.8 |
| Cambrian-8B [47] | 7.9 | 30.9 | 27.9 | 14.6 | 15.5 | 19.4 | 65.0 | 70.1 | 59.3 | 79.1 | 86.0 | 72.8 |
| InternVL2-8B [44] | 8.5 | 27.0 | 22.1 | 13.3 | 7.5 | 15.7 | 55.5 | 61.0 | 60.4 | 74.7 | 64.3 | 63.5 |
| Qwen2-VL-7B [4] | 10.2 | 25.2 | 36.1 | 17.3 | 17.4 | 21.2 | 55.5 | 61.0 | 60.4 | 74.7 | 64.3 | 63.5 |
| VILA1.5-13B [24] | 11.0 | 34.1 | 36.8 | 16.8 | 19.4 | 23.6 | 61.8 | 63.2 | 60.1 | 73.9 | 59.8 | 63.7 |
| LLaMA-3.2-11B [11] | 22.9 | 32.5 | 46.2 | 17.6 | 15.7 | 27.0 | 66.2 | 75.0 | 78.6 | 89.6 | 79.9 | 78.3 |

Table 2: Performance of VLMs on our AGMMU OEQs and MCQs. Our evaluation set poses great challenges to existing large VLMs, including closed-source and open-sourced models.

## 4.1 Baselines

We include a broad range of state-of-the-art VLMs to ensure robust comparisons. For open-sourced models, we prioritize architectures with comparable parameter numbers for a fair and meaningful comparison. Moreover, we include the most recent, largest, and highest-performing checkpoints up to the date of our evaluation questions, including LLaMA-3.2 [11], LLaVA-1.5 [25], LLaVA-NeXT [26], LLaVA-OneVision [22], Cambrian-1 [47], InternVL-2 [44], Qwen-VL [4], and VILA [24], as well as proprietary models such as GPT-o4-mini [1], Claude-3.5 [41], and Gemini1.5-Pro [42]. Detailed configurations and model settings are provided in the supplementary material.

## 4.2 Evaluation Metrics

We evaluate the models on multiple-choice questions (MCQs) and open-ended questions (OEQs). To ensure a robust and fair evaluation, we have developed a systematic approach to minimize potential variations in model responses due to intermediate generations or formatting inconsistencies.

**For MCQs**, we report accuracy as the primary evaluation metric. We score model response with string pre-processing and matching the predicted letter with the ground truth letter.

**For OEQs**, we implement the LLM-as-judge methodology using GPT-4.1 to grade the answers. The prompts and style are initialized from SimpleQA [51] setting, but adapted for AGMMU scenarios. The grading is divided into two categories: (1) short-form responses and (2) multi-statement responses. The **short-form responses** correspond to the questions from *pest identification*, *disease identification*, and *species identification*, normally containing several words, and are evaluated directly against the correct answer. We assign the grades of "correct", "incorrect", "partially correct", and "not attempted," then use the harmonic mean to calculate the final grade of "Correct/Total" and "Correct/(Total − Partially Correct)", similar to SimpleQA. The **multi-statement responses**, on the other hand, are for *management instructions* and *symptom descriptions* questions. These categories usually consist of multiple unique and standalone statements: On average, management instruction responses contain 2.7 factual statements, and symptom/visual description responses contain 1.8 factual statements. Therefore, we have our LLM judge (a) divide both predicted and ground-truth responses into individual statements; and (b) grade each of them as a short-form response; then (c) normalize the grades according to the number of statements per question to calculate the final harmonic mean score as short-form responses.

As additional context, we emphasize the importance of a partially correct category to capture the nuances of agricultural knowledge. For example, (1) *Taxonomic Hierarchy*: In disease identification,

guessing "fungus" when the correct answer is "black jelly fungus" is considered partially correct; (2) *Semantic Granularity*: In management instructions, suggesting "pesticide" when the correct answer is "spray insecticidal soap, carbaryl, or spinosad" is partially correct because pesticide includes the correct answer despite a specificity difference. Compared with SimpleQA, whose metric implicitly encourages "not attempted," we exclude it as the agricultural questions from farmers commonly emphasize instantaneity.

### 4.3 Zero-shot Evaluation Results

We present a comprehensive zero-shot comparison of various VLMs in Table 2. More visualizations are presented in the supplementary material.

**Very Challenging for Existing Models.** AGMMU proves to be a very challenging benchmark across all models evaluated, and even the most advanced systems achieve moderate performance levels. Such challenges are notable, especially on open-ended OEQs, where models have to recall the knowledge precisely without relying on any pre-input options. These observations also reveal the necessity of improving the agricultural expertise of VLMs and mixing agricultural data for VLM training, where our AGBASE could be helpful.

**Performance Across Question Types.** On MCQs, the models can rely on options, so it is significantly easier than OEQs. The level of >70% accuracy is also on par with previous agricultural benchmarks [28, 55] in MCQs. Therefore, we primarily advocate for the more challenging OEQ setting, which is also more aligned with a VLM in the real world. As observed in Table 2, the *management instruction* and *symptom/visual description* types of questions pose more challenges, due to their multi-statement and long-response nature and the need for more reasoning. However, there is significant variation between models as to what tasks are the most difficult, suggesting that specialized agricultural knowledge and visual understanding capabilities are not uniformly distributed across model architectures.

**Closed- and Open-sourced Models.** We observe a distinct performance gap between closed-source and open-sourced models, where closed-source models generally demonstrate superior performance. Among all models, Gemini 2.5 Pro leads in all short-form categories, Gemini 1.5 Pro in management instruction, and GPT-4o in symptom description. For open-sourced models, LLaMA-3.2 and VILA1.5 are the overall best performer across all subdomains. On average, open-sourced models trail behind their closed-source counterparts by more than 10 to 20%, likely due to the lack of substantial agricultural data during the large-scale training phases.

### 4.4 Reliability of LLM Judge

The reliability of the LLM judge is essential for meaningful evaluation, so we ensure its robustness through rigorous prompts and validate its performance via a human verification study.

| Category | Correct / Total |
|---|---|
| Disease/Issue Identification | 9 / 10 |
| Insect/Pest | 9 / 10 |
| Species | 10 / 10 |
| Management Instructions | 10 / 10 |
| Symptom/Visual Description | 9 / 10 |

Table 3: Agreement between GPT-4.1 Judge and Human Expert on 50 randomly sampled model responses. The scores indicate a high level of reliability.

**Empirical Human Agreement.** To quantify the alignment between the LLM judge and human assessment, we conducted an empirical study. We randomly selected 50 responses generated by our fine-tuned LLaVA model, with 10 samples from each of the five categories. A human expert then manually graded these responses. As shown in Table 3, the judgments from GPT-4.1 exhibit a high degree of alignment with human evaluations.

**Nature of Disagreements.** We observed that the few mismatches between the LLM judge and the human grader were primarily borderline cases. In these instances, GPT-4.1 typically assigned a "partially correct" label to an answer that the human grader deemed "incorrect." For example, when the ground truth was "leaf-spotting fungi or bacteria," the judge rated the prediction "bacterial wilt disease" as partially correct. This assessment is defensible due to the shared keyword ("bacteria") and overlapping symptoms, indicating it is not a significant failure of reasoning.

**Robust Evaluation Metric.** Crucially, our scoring metric is designed to be robust against such borderline cases. As detailed in Sec. 4.2, credit is awarded only for answers that are fully correct.

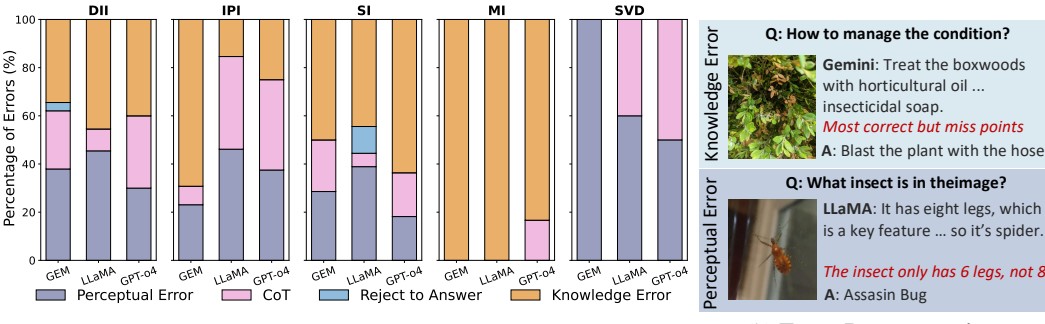

(a) **Error Analysis**.  (b) **Error Demonstration**.

Figure 4: (a) The most common errors made by VLMs are knowledge errors. CoT represents samples that are originally wrong due to false reasoning, but corrected with chain-of-thought prompting. IPI, DII, SVD, SI, and MI are short for five question types as explained in Sec 3.2. (b) We show two common evaluation errors, including the lack of knowledge to answer the question (*top*), and the wrong perception of the image (*bottom*).

Consequently, these "partially correct" labels assigned by the LLM judge do not inflate the final performance scores.

### 4.5 Error Analysis

To achieve an in-depth understanding of the internal procedures of VLMs in answering AGMMU questions, we analyze the types of error that occurred during the evaluation. We select 171 wrong evaluation samples and ask the VLMs to provide the chain-of-thought (CoT) reasoning before providing the final answer, and judge the error types with human evaluators. Fig. 4a demonstrates quantitative results, and Fig. 4b shows the qualitative results of the most common knowledge and perceptual errors.

**Perceptual Error.** A perceptual error occurs when the VLM fails to recognize a primary visual characteristic or gives a direct incorrect description of a visual characteristic as in Fig. 4b. This category highlights limitations in the visual understanding capabilities of VLMs. Perceptual errors constitute 34.3% (Gemini), 37.5% (LLaMA), and 22.2% (GPT-o4) of total errors.

**Knowledge Error.** A knowledge error occurs when the VLM does not have the essential knowledge to reach the correct answer. In these cases, the model may identify relevant visual features and reinforce it, but without reaching the correct answer. For example, in management questions, the model might give vague responses that do not engage with the particular nuances of the given situation (Fig. 4b). Knowledge errors constitute the highest proportion of errors for all Gemini (48.6%), LLaMA (42.9%), and GPT-o4 (51.1%).

**Chain-of-Thought (CoT).** This category includes instances where the VLM initially provides an incorrect response but corrects itself after being asked to explain its reasoning. The proportion of CoT errors underscores the importance of reasoning processes for accurate performance on our benchmark. Specifically, CoT accounts for 15.7% (Gemini), 16.1% (LLaMA), and 26.7% (GPT-o4) of total errors, indicating that these models benefit significantly from explicit reasoning steps.

**Reject to Answer.** Occasionally, VLMs refuse to answer a question entirely, reflecting uncertainty or lack of confidence in their predictions. This occurs rarely, constituting 1.4% (Gemini) and 3.6% (LLaMA), with no instances observed in GPT-o4.

The differing error patterns across Gemini, LLaMA, and GPT-o4 reveal distinct model behaviors. Perceptual ability forms the foundation: models like LLaMA, with weaker visual grounding, show more perceptual errors—often hallucinating features to match textual cues. In contrast, Gemini accurately identifies main visual elements without hallucination but frequently makes knowledge errors, indicating gaps in domain understanding despite solid perception. GPT-o4, with strong perception and knowledge, rarely makes blatant errors, but its reasoning steps can still misfire.

### 4.6 Finetuning Evaluation Results

**Experiment Setting.** We finetune both Qwen2.5-VL [50] and LLaVA-1.5-7B [25] using our AG-BASE. Our training follows the standard practice of LLaVA by combining its instruction-tuning

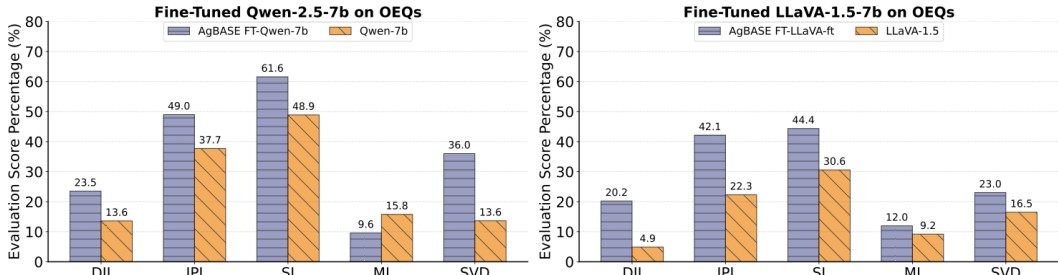

Figure 5: Fine-tuning on AGBASE boosts the agricultural knowledge understanding for both Qwen2.5-VL and LLaVA-1.5, indicating the potential of our development set. IPI, DII, SVD, SI, and MI are the short for the five question types as explained in Sec. 3.2.

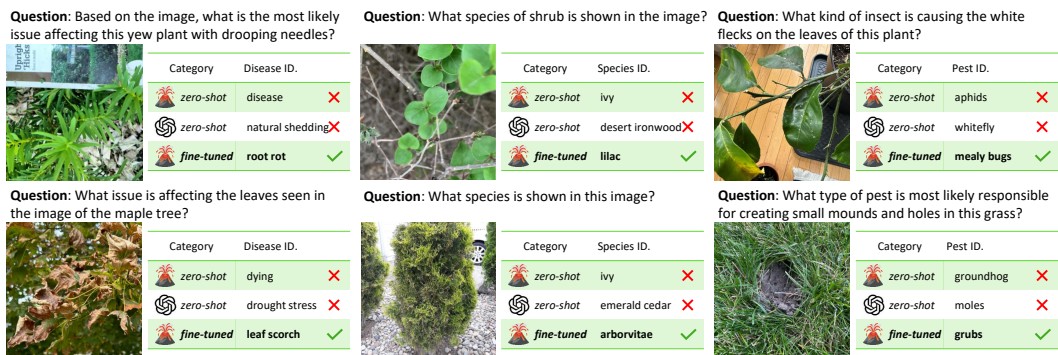

Figure 6: The effectiveness of AGBASE fine-tuning on OEQ examples. After simple fine-tuning, the LLaVA model can accurately identify issues that GPT and the original model fail to recognize in zero-shot scenarios.

set and AGBASE together, where the instruction-tuning set of LLaVA is critical for instruction-following abilities to correctly answer MCQs. Our training lasts two epochs using LoRA adapters [20], a learning rate of 2e-4, a weight decay of 0, and a batch size 16. The fine-tuning process takes approximately 12 hours on 2×A6000 GPUs.

**Analysis.** Fig. 5 and Fig. 6 demonstrate the effectiveness of fine-tuning VLMs using our AGBASE on the more challenging open-ended questions. Fig. 5 shows that fine-tuning with our knowledge base significantly improves the model's capability of understanding the images and correctly responding with agriculture knowledge. This performance boost highlights the effectiveness of our large-scale knowledge base in improving VLMs and the necessity of collecting agricultural-related data for future VLMs. Meanwhile, the smaller improvement on management instruction and symptom description questions indicates the complexity of multi-statement questions, motivating the need for more data collection and better model training than simple fine-tuning.

## 5 Conclusion

In this work, we introduced AGMMU, the first benchmark for evaluating vision-language models (VLMs) in agriculture, a field that requires precise visual interpretation and expert knowledge. Our dataset spans core agricultural tasks, including symptom recognition, species and pest identification, and management instructions, and is created from more than 116,231 real-world expert-user interactions. Through a three-step curation framework, we build a high-quality AGMMU evaluation set, demonstrating a great challenge for current VLMs. To support model development, we introduce AGBASE, a training set of 57,079 knowledge entries aimed at improving model accuracy. We hope that the combination of AGMMU and AGBASE can support the community in evaluating and developing stronger knowledge-intensive VLMs.

## Acknowledgments

We are grateful to the AskExtension team for providing the dataset. This work was supported in part by the NIFA Award 2020-67021-32799, NSF under Grants 2106825 and 2519216, the DARPA Young Faculty Award, the ONR Grant N00014-26-1-2099, and the Center for Digital Agriculture at the University of Illinois. This work used computational resources, including the NCSA Delta and DeltaAI supercomputers through allocations CIS230012, CIS230013, and CIS240133 from the Advanced Cyberinfrastructure Coordination Ecosystem: Services & Support (ACCESS) program, as well as the TACC Frontera supercomputer, Amazon Web Services (AWS), and OpenAI API through the National Artificial Intelligence Research Resource (NAIRR) Pilot.

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

# Contents

## A  Evaluated Large Vision Language Models

Our evaluation and analysis are conducted mainly on the group of models listed in Table 2 in the main paper. We have chosen models such that they cover most of the popular and best-performing methods used by recent multimodal understanding work. In this part, we discuss all the models we have used in our experiments and explain their evaluation details, the public checkpoints we have chosen, and display the prompts we used to adapt the model to our datasets.

During evaluation, we chose to follow the standard prompt provided by the authors whenever possible for multiple-choice and short-answer questions. When the prompt is not provided for the model, we select a custom prompt that is created through several iterations of prompt engineering to select the one that produces the most effective results. The images are always included as the prefix.

**Proprietary Models.**  We used six proprietary models in our evaluation: GPT-o4-mini [1], Gemini 1.5 Pro [34], Claude 3 Haiku [41], GPT-4o, Claude 3.5, and Gemini 2.5 Pro. Below we note the model API version used for evaluation.

- GPT-o4-mini: May 13-15, 2025; October 2-3, 2025.
- Gemini 1.5 Pro: November 1-13, 2024; October 2-3, 2025.
- Claude 3 Haiku: November 13-14, 2024; October 2-3, 2025.
- GPT-4o: July 25-28, 2025; October 2-3, 2025.
- Claude 3.5: July 25-28, 2025; October 2-3, 2025.
- Gemini 2.5 Pro: July 25-28, 2025; October 2-3, 2025.

**Cambrian-1**  [47]. Cambrian-1 is a recent state-of-the-art model that excels at visual-centric tasks. This model explores combinations of vision encoders, text and image integration techniques, and instruction tuning strategies. We use the official implementation and checkpoint[1] with a LLaMA3-8B-Instruct LLM backbone model in our evaluation.

**InternVL2**  [44]. InternVL scales up the vision foundation model while aligning it with the backbone LLM, and is trained on web-scale image-text data to achieve strong performance across a variety of vision-centric tasks. We use the official implementation and checkpoint[2] with the InternViT-300M-448px vision backbone and Internlm2.5-7B-chat language backbone in our evaluation.

---

[1] https://github.com/cambrian-mllm/cambrian
[2] https://huggingface.co/OpenGVLab/InternVL2-8B

**LLaMA-3.2** [11]. LLaMA-3.2 is the first collection of multimodal large language model from the LLaMA family that was previously text-only. The integration of vision involves utilizing cross-attention layers and a pre-trained vision encoder that feeds directly into the text-processor. The model follows a commonly used training recipe that includes pretraining on noisy image-text pairs and then high-quality knowledge enhanced pairs. Notably, the language-model parameters were frozen during the training of alignment of image and text to retain strong text-only capabilities. We use the official implementation and checkpoint[3] that uses a LLaMA-3.1 text-only language backbone in our evaluation. When evaluating the model, we choose to use a custom prompt since no standard prompt is provided.

**LLaVA-NeXT** [26]. LLaVA-NeXT expands on LLaVA by using the same instruction tuning method to give the model the ability to process and reason about multi-images, multi-grames, and multi-views. We use the official implementation and checkpoint[4] with LLaMA-3-8B Instruct as the language backbone in our evaluation.

**LLaVA-OneVision** [22]. LLaVA-OneVision builds on LLaVA-NeXT with the capability to analyze single images, multi-images, and video scenarios. Most impressively, it allows for video understanding through task transfer from images but this is not explored in our evaluation. We use the official implementation and checkpoint[5] that uses a base architecture consisting of SigLIP-SO400M-Patch14-384 and Qwen2-7b in our evaluation.

**LLaVA-1.5-7B / LLaVA-1.5-13B** [25]. LLaVA introduces the idea of instruction tuning a multimodal model with GPT-4 generated instruction-following data for associated images. This gives it the ability to achieve impressive abilities to act as an instruction-following general agent. We use the official implementation and checkpoints[67] with a CLIP ViT-L/14 vision backbone and Vicuna1.5-7B / Vicuna1.5-13B in our evaluation.

**Qwen-VL-7B** [4]. Qwen-VL is a large vision language model that has the ability to perform various vision-language tasks including image captioning, visual grounding and more, not only limited to question answering. This model is multi-lingual in Chinese and English and was pre-trained using an interleaved image-text technique. We use the official implementation and checkpoint[8] that uses Qwen-7B as the language backbone and CLIP ViT bigG/14 as the vision encoder in our evaluation.

**VILA1.5-13B** [24]. VILA is trained using an enhanced pre-training method that involves interleaved visual language data. Additionally, during the supervised fine-tuning stage, the data includes text-only instruction data to help the model retain strong text-only capabilities. We use the official implementation and checkpoint[9] with a LLaMA3-8B LLM backbone and SigLIP-SO400M-Patch14-384 vision encoder in our evaluation.

## B    Dataset Curation Details

This section outlines the multi-stage curation pipeline of AGMMU and describes the prompts designed for each question type and subdomain.

### B.1    Stage 1: Question Categorization

In the first step, we employ the Llama-70B model [11] to categorize questions into predefined agriculture subdomains while identifying the primary living entity affected by the query. Our systematically crafted prompt (Figure A) guides the model to extract the most specific living entity mentioned, such as "apple tree" or "honeybee," or to assign "none" when the entity is unclear or absent.

The subdomains include *Disease, Weeds/Invasive Plant Management, Insect/Pest Control, Growing Advice, Environmental Stress, Nutrient Deficiency, Generic Identification*, and *Other*. Each sub-

---

[3] https://huggingface.co/meta-llama/Llama-3.2-11B-Vision
[4] https://huggingface.co/llava-hf/llama3-llava-next-8b-hf
[5] https://huggingface.co/llava-hf/llava-onevision-qwen2-7b-ov-hf
[6] https://huggingface.co/llava-hf/llava-1.5-7b-hf
[7] https://huggingface.co/llava-hf/llava-1.5-13b-hf
[8] https://huggingface.co/Qwen/Qwen-VL
[9] https://github.com/NVlabs/VILA

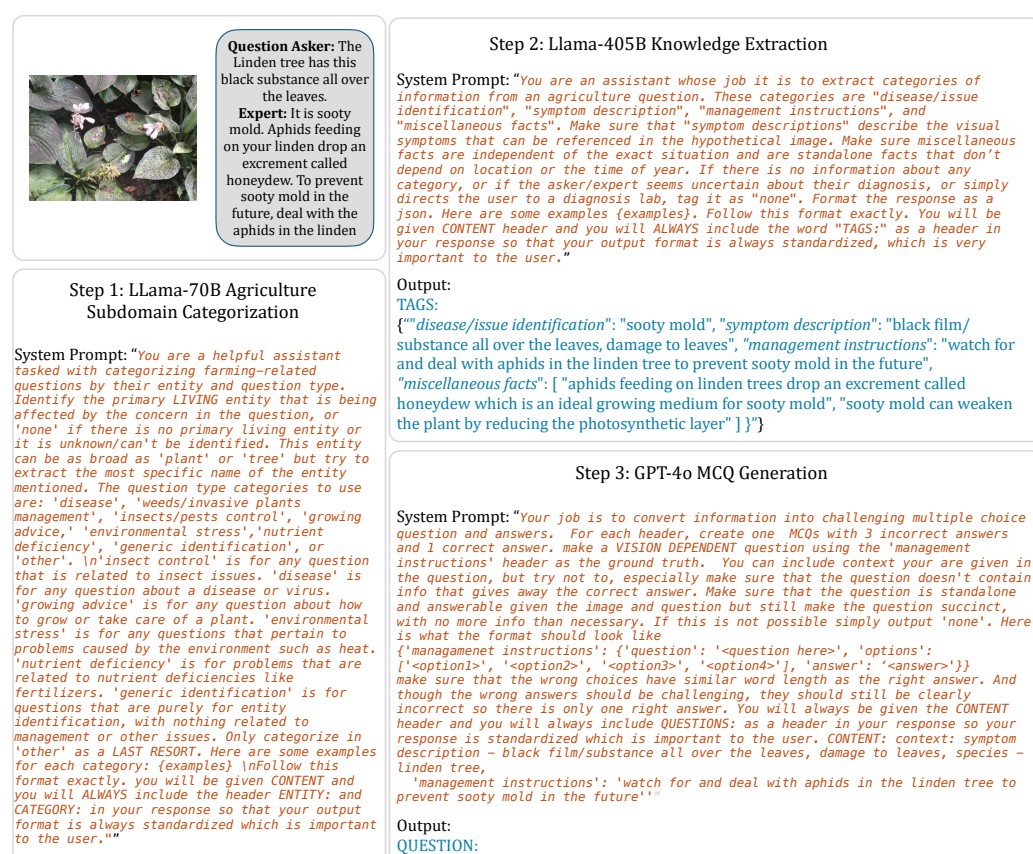

Figure A: Prompts used in different stages of our data curation pipeline.

domain is succinctly defined within the prompt, with illustrative examples provided in Figure B to address ambiguous or edge-case scenarios. The prompt enforces a standardized output format, ensuring consistency with the inclusion of "ENTITY:" and "CATEGORY:" headers.

To enhance robustness, the prompt includes examples of complex or overlapping cases, ensuring accurate classification even for questions that span multiple subdomains or lack explicit details. By embedding these clarifications, the design supports reliable categorization across diverse agricultural contexts.

## B.2 Stage 2: Information Extraction

In the second step, we design prompts to extract granular categories of information from agricultural questions. These categories are tailored to the specific subdomain identified in Step 1, ensuring that the extracted information is both relevant and actionable.

**Weeds/Invasive Plants Management.** For the "weeds/invasive plants management" subdomain, the extraction focuses on: (1) *Image Description*, visual characteristics of the weed or invasive plant, (2) *Management Instructions*, actionable strategies for control, and (3) *Miscellaneous Facts*, contextual expert insights. The name of the weed itself is already extracted in Step 1. This ensures that the emphasis remains on descriptions, actionable measures, and expert knowledge.

**Insects/Pests Control.** For this subdomain, the categories include: (1) *Insect/Pest*, identifying the pest in focus, (2) *Image Description*, visual traits of the pest or evidence of damage, (3) *Management Instructions*, guidance for mitigation, and (4) *Miscellaneous Facts*, contextual expert insights. The primary plant affected, if exists, is identified in Step 1, thus this step concentrates on pest-specific details, such as visual features or damage patterns, and the corresponding management strategies.

# Agriculture Domain + Species Extraction (step 1) Examples

CONTENT:
Best ways to treat Azaleas and Mt. Laurel with a Lace infestation. #875156

*Question Asker:*
 My Azalea shrubs and Mt. Laurel are infested with Lace.  I have sprayed
them but I am not sure if I spayed them adequately.  ISome of the shrubs are
tall, and I will need a ladder to reach the top.

Is it too late to apply a liquid application around the base? How often
should my shrubs be treated? Should I hire a private company?

any suggestions would be much appreciated.
*Expert:*
Can you share a photo or two of what you are seeing? You can attach them
directly to this reply. First off, stop spraying, and let us know what you
are using. You wouldn't see damaged leaves recover, you would just see any
new leaves look healthy. You can also burn leaves of plants (or affect non-
target insects and plants when spray applications are made when the weather
is hot or windy.  Lace bugs tend to be worse in landscapes where azaleas
are planted in full sun (which stresses them) and where pesticides are
regularly used. In healthier landscapes with little or no pesticide use and
an abundance of different plants, their populations are kept in check by
beneficial insects. That is the ideal goal.  Azalea can get lace bugs that
are specific to azalea that you can learn about here: <link> there are also
different lace bugs that are specific to Rhododendrons and Japanese
Andromeda as well but we don't see them on Mountain Laurel.  Let us see
what your concerns are on the Mountain Laurel and we will assist. The most
common problems that those shrubs have tend to be holes in the leaves
(Shothole, which can look like insect chewing but is not, and is cosmetic
and no chemical controls are recommended) and bark scale insects, which
would looks like white flocking along the limbs.  Systemic soil drenches
containing imidacloprid (a type of neoniconoid) have been found to be
damaging to pollinators. In 2016, the Maryland Pollinator Protection Act
was passed, which prohibits homeowners from applying them. Only
professional, licensed applicators may do so. For this reason, applications
would only be a last resort, and there are many other, less toxic, more
environmentally friendly ways to deal with many pests. Here is a page that
explains more: <link>

ENTITY:
azalea, mt. Laurel
CATEGORY
insects/pests control

CONTENT:
Large brown spots on bush bean leaves #873890

*Question Asker:*
 Hi,

I'm wondering what these brown patches on my green beans are and if
there's anything I should do to stop/prevent further issues.

Thanks!
*Expert:*
This looks like abiotic damage, which means it was caused by environmental
factors and not a pest or disease. In this case, it looks like sunscorch
(also called sunscald or just "scorch"), which is essentially sunburn.
Plants with reduced air circulation, such as being crowded or growing near
a wall, solid fence, or near heat-reflective pavement or stones can be
more vulnerable to scorch, but even well-spaced and unobstructed plants
can still develop it. Beans can be among the more vulnerable veggies to
scorch.  Fortunately, mild scorch in beans generally does not affect
yield. You can keep monitoring the plants for watering needs, feeling the
soil a few inches down and watering if it becomes somewhat dry to the
touch, but no other intervention is needed. Floating row cover and insect
mesh netting can serve as a shade cloth of sorts (even if not needed for
their pest-excluding or frost-shielding properties) if a full sun exposure
is proving to be too much for certain plants, but we'd expect these plants
will grow out of it well enough on their own. (Injured leaves cannot heal,
but new foliage should emerge normally.)

*Question Asker:*
Glad to hear this. Thanks

*Expert:*
You're welcome!

ENTITY:
green beens
CATEGORY
environmental stress

CONTENT:
Dead Grass #829812

*Question Asker:*
 Hi There,

Last summer was hard on my grass with most of it dying, particularly in
full sun areas.  I'm left with some dead patches but mostly bare dirt.
I'm interested in doing a no-mow grass on my slope, regular on the flat
yard, and am looking for recommendations on if i should sod/seed and what
varietals and extra care steps (fertlizer, watering times, etc.) you
might recommend. Thank you kindly!

*Expert:*
No mow options can exceed city ordinances. Because city
ordinances sometimes limit what can be planted on boulevards, you might
want to check that first.  It would also be good to get a soil test to
see what plants are a good match to your soil. See: https://
soiltest.cfans.umn.edu/  The steepness of some of the area suggests that
you also need erosion control for the area.  The following websites offer
some ideas that may help you decide.  Whatever you chose, a deep rooted
planting is better for this area that shallow rooted plants like grass.
1.  For steep slopes see page 34.  <link> 2. For landscape design see:
 <link> 3.  For native prairie plants that require no fertilizer or
watering see: <link>  4.  For low growing ground covers see:<link> 5.
Also see: <link>   You could also take a trip to the Minnesota Arboretum
in Chanhassen and see some examples of plants that may interest you.

ENTITY:
grass
CATEGORY
growing advice

CONTENT:
What is this plant? #874057

*Question Asker:*
 I originally got this as a stray seedling with a peony plant I purchased
at a local nursery.  I potted it out of curiosity.  It's grown into a
lovely good sized plant.  Can you tell me what this is?  Thank you for
your help.
*Expert:*
Hello, happy to help. I suspect it may be a weed but I'd be happy to
continue working with you to identify it. Could you send a photo of its
flower and what month it bloomed when that happens? Though the question
may look closed, when you add a reply, it will reopen and notify me.
Thanks!

*Question Asker:*
So far there hasn't been a hint of blossom or flower.  Below are pics from
just now.  The largest leaf is now 7.5".I have not seen anything like this
growing wild in our area.  I live in rural Isanti county.  Sandy soil
country. Thank you so much for your help

*Expert:*
Hello, It's burdock an invasive weed. The common burdock can be found
everywhere in Minnesota but there are three varieties and all of them are
invasive and should be eradicated. Here is information about all three
types from Minnesota Wildflowers. You could cut one of the stems to see if
it's hollow or not. If not hollow, it is the likely newer variety called
Actium lappa.  Good-luck!

*Question Asker:*
Thank you so much for researching this for me.  The leaves do look similar
to the Great Burdoch.  Leaves on the other 2 are too pointy.  I don't see
any branchy stem coming up for flower buds.  I got this seedling in
April.Perhaps this matures late summer?  Being as it is contained in a pot
on my patio I will let it mature to see what it does.  Should be
interesting.  Thank you again for naming my Mystery Plant and letting me
know I shouldn't plant it in the garden!

ENTITY:
burdock
CATEGORY
weed/invasive plants management

CONTENT:
Question about freezer jam #875023

*Question Asker:*
 Hello. I'd like to make both a strawberry and strawberry rhubarb freezer
jam, however, for health reasons, I'd prefer to use raw honey in place of
sugar.

I'm curious--can I substitute honey for sugar in any freezer jam recipe,
and, if so, how much? Also wondered if you had any recipe suggestions in
this vein.

Secondly, I have found recipes that already call for honey in lieu of
sugar, if I was to use these or make my own substitution and use Suregel,
is it safe to let the jam sit out at room temperature for the 24 hours
requied when using Suregel?

Thank you for your time.

*Expert:*
Hi, As per the National Center for Home Food Preservation and USDA, Corn
syrup and honey may be used to replace part of the sugar in recipes, but
too much will mask the fruit flavor and alter the gel structure. Use
tested recipes for replacing sugar with honey and corn syrup. (<link>) If
you are trying to reduce sugar, please know that honey is also pure sugar,
just from a different source — so simply substituting this is not a
solution to that challenge.  There is information in the above link that
does talk about making jams/jellies with reduced sugar — one option is
using a "low-methoxyl pectin", which the brand name is Pamona, another
option you may want to try.   You can substitute honey in Suregel products
and it is safe to leave out for 24 hours when canned.    I hope I have
answered all of your questions, if not, please respond with further
questions. Thank you,

ENTITY:
strawberry
CATEGORY

Figure B: Examples included in prompt during the agriculture domain categorization (step 1).

Knowledge Type Extraction (step 2) Examples

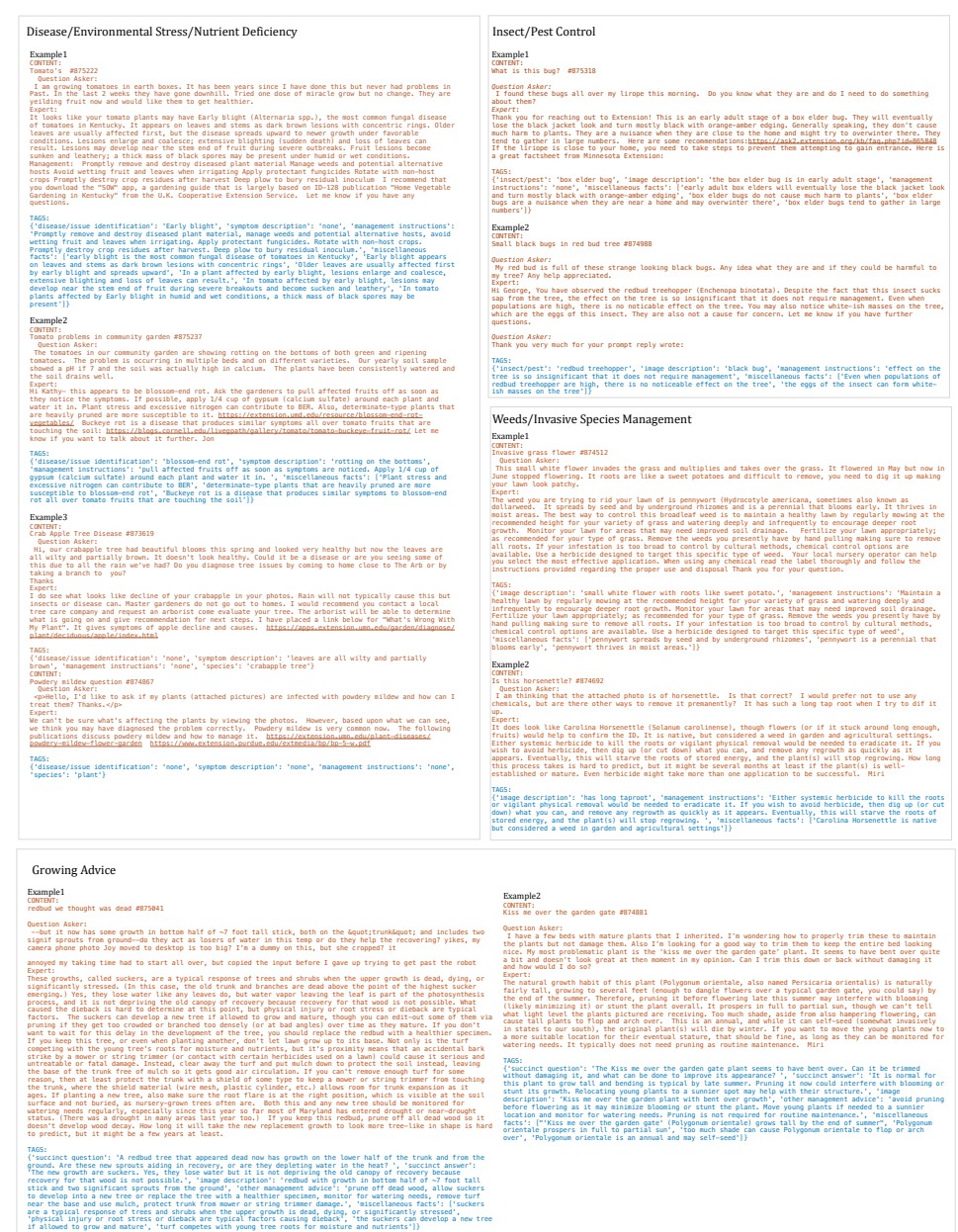

Figure C: Examples included in prompt during the knowledge extraction (step 2) based on agriculture subdomain type.

Figure D: Prompts used for each subdomain during knowledge extraction (step 2).

**Nutrient Deficiency, Disease, Environmental Stress.** For these subdomains, we group them due to shared characteristics. The extracted categories are: (1) *Disease/Issue Identification*, specifying the underlying cause, (2) *Symptom Description*, observable signs such as discoloration or stunted growth, (3) *Management Instructions*, remediation or prevention strategies, and (4) *Miscellaneous Facts*, contextual expert insights. These subdomains are defined by their symptomatic presentation, the underlying conditions, and the need for targeted management interventions.

**Growing Advice.** For this subdomain, the variability in question structure necessitates tailored extractions: (1) *Succinct Question*, a concise reformulation of the user query, (2) *Succinct Answer*, a precise response to the query, (3) *Image Description*, any relevant visual details, and (4) *Miscellaneous Facts*, contextual expert insights.

Importantly, besides distinguishing the extraction types, we also put different examples of pre-made knowledge extraction into the prompt, see Figure C. Prompts given to the model for each subdomain can be seen in Figure D.

The *Miscellaneous Facts* category is extracted across all subdomains but is not directly used in subsequent steps. Instead, it captures standalone expert information that can contextualize a user's issue.

To optimize extraction accuracy, we distinguish between "Symptom Description" (used for nutrient deficiency, disease, and environmental stress) and "Image Description" (used for weeds/invasive plants and insects/pests). While these serve a similar purpose—capturing observable or visual details—they are unified under the term "Symptom/Visual Description" in subsequent steps to maintain consistency.

# Short Answer OEQ Grading Prompt

Your job is to grade student answers from the agriculture and biology domain. Your job is to look at a
question, a gold target, and a predicted answer, and then assign a grade of either ['CORRECT', 'INCORRECT',
'NOT ATTEMPTED', 'PARTIALLY CORRECT'].

First, I will give examples of each grade, and then you will grade a new example. {examples}

Remember the following key points:
    — a statement should be AT LEAST partially correct if the predicted answer is a subcategory of the
        gold target or the gold target is a subcategory of the predicted answer
    — a statement is always partially correct if it has ANY overlap in content with the target

Grade the predicted answer of this new question as one of:
A: CORRECT
B: INCORRECT
C: NOT_ATTEMPTED
D: PARTIALLY CORRECT

Question: {question}
Gold Target: {target}
Predicted Answer: {predicted_answer}

Just return the letters "A", "B", "C", or "D", with no text around it.

Figure E: Grading prompt for our LLM-as-judge on short-answer OEQ.

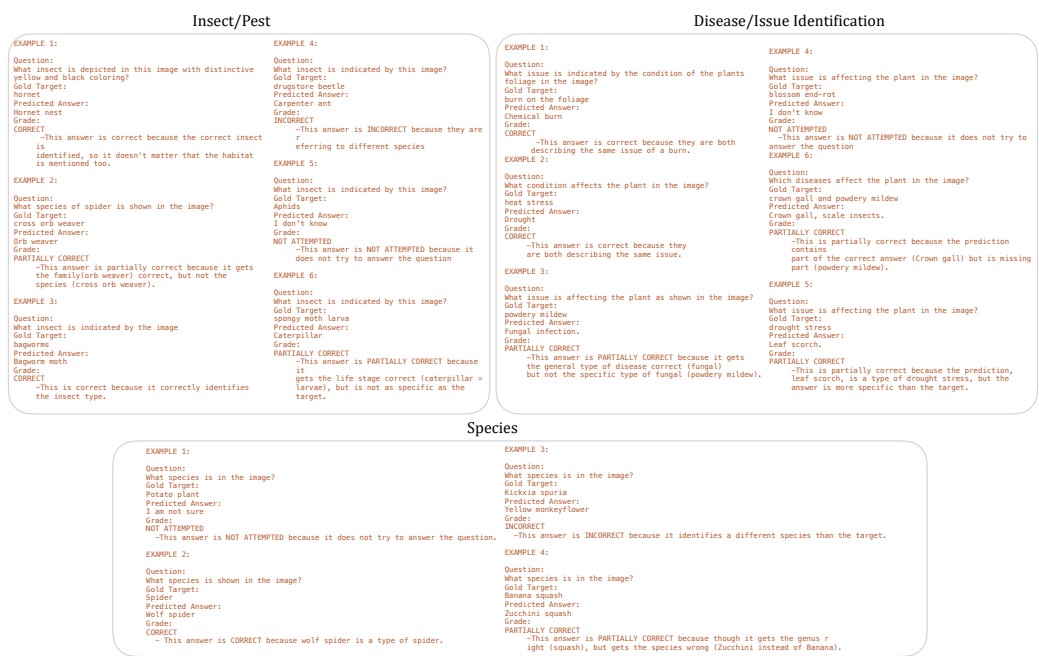

Figure F: Unique examples included for short-answer categories added to the grading prompt for
our LLM-as-judge on short answer OEQ.

# Multi-Statement OEQ Grading Prompt

```
Your job is to grade student answers from the agriculture and biology domain. Your job is to look at a
question, a gold target, and a predicted answer, and then assign grades to each statement in the response
of ['correct','partially correct', 'incorrect', 'missing', 'irrelevant'].
        —Correct is assigned to statements from the predicted answer that fully semantically map to a
         statement in the gold target.
        —Partially correct is assigned to statements which partially semantically map to a statement in the
         gold target.
        —Incorrect is assigned to statements from the predicted answer that directly semantically contradict
         a statement in the gold target.
        —Missing is assigned to statements in the gold target which haven't been mapped within correct,
         partially correct, or incorrect.
        —Irrelevant is assigned to statements in the predicted answer that directly respond to to the
         question but do not contradict nor correspond in any way to statements in the gold target.
         EACH STATEMENT IN THE GOLD TARGET AND PREDICTED ANSWER SHOULD BE ASSIGNED TO EXACTLY ONE OF THESE
         CATEGORIES.

        Here are examples of correctly graded statements: {examples} Remember the following key points:
            —a statement is always partially correct if it has ANY overlap in content with the target
            —If there are multiple statements that match a gold target statement, only match it with the
             best one, and put the rest in irrelevant.
            —Ignore any statement that are not directly attempting to answer the question, and do not
             assign them to any of the categories, not even irrelevant. {specific_instructions}

        Question: {question}Gold Target: {expected} Predicted Answer: {actual} Follow the format of the examples
        exactly. Output only a parsable json with no additional text, special characters or formatting mistakes.
```

Figure G: Prompt for categorizing statements in our LLM-as-judge on multi-sentence (long-answer) OEQ.

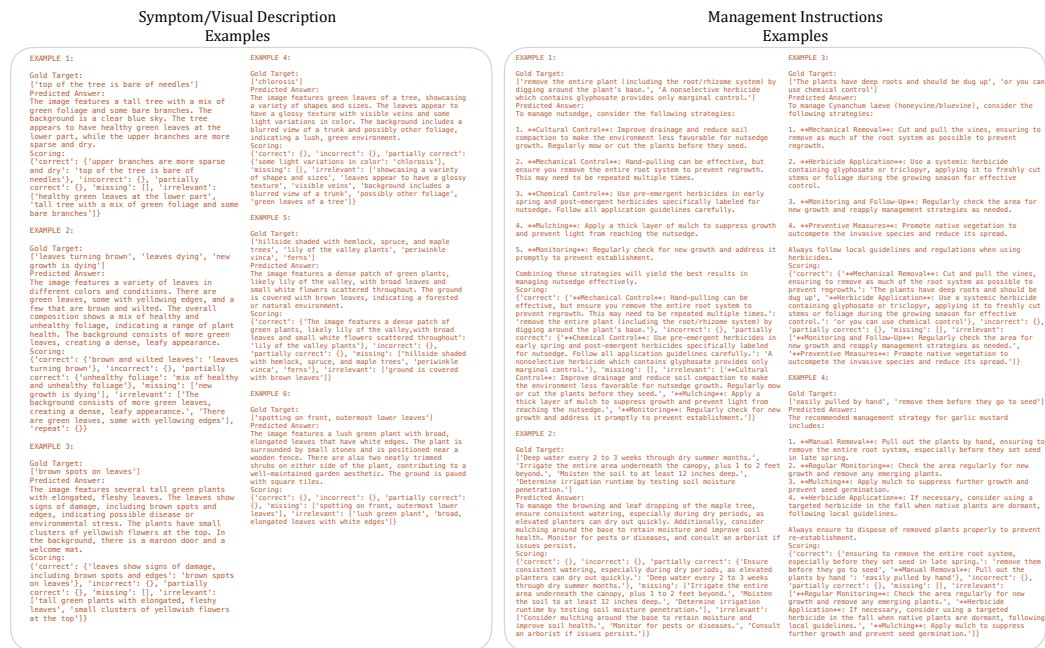

Figure H: Unique examples included for multi-statement categories added to the grading prompt for our LLM-as-judge on multi-sentence (long-answer) OEQ.

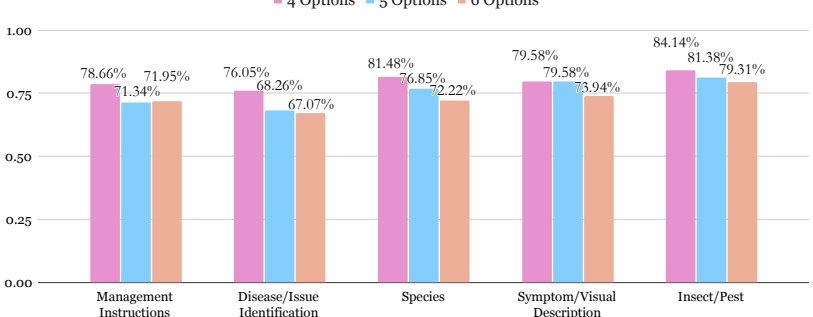

Figure I: **GPT-4o accuracy with increasing MCQ options.** Model performance on MCQs across different categories, comparing accuracy scores when varying the number of answer options (4, 5, and 6). We observe a 5-10% difference in accuracy across categories between the 4-option and 6-option configurations, with performance generally decreasing as the number of options increases.

## B.3    Stage 3: Question Generation

In the final step, the extracted agricultural facts are transformed into evaluative question-answer (QA) pairs, comprising multiple-choice questions (MCQs) and open-ended questions (OEQs) generated using GPT-4o. To enhance relevance, we exclude two knowledge types: (1) *Growing Advice*, as image content often lacks direct correlation with the user's issue, and (2) *Miscellaneous Facts*, since these provide general context but do not directly relate to the user's image. This refinement narrows the scope to five key knowledge types for downstream processing, including *Disease/Issue Identification*, *Symptom/Visual Description*, *Management Instructions*, *Insect/Pest*, and *Species*.

To ensure clarity and relevance, we employ a standardized prompt structure (see Figure A tailored to each knowledge type. While the core structure remains consistent, the phrasing explicitly references the specific knowledge type being addressed. This targeted design allows the prompts to focus on generating well-contextualized and relevant questions. For added precision, the prompts incorporate contextual details where applicable: (1) For *species-related questions*, only symptom/visual description information is referenced, ensuring the focus remains on observable traits, and (2) for *symptom/visual-related questions*, species information is used to provide context, helping to ground the questions in specific agricultural scenarios.

This contextualization ensures that the generated questions integrate both user-provided information and extracted context seamlessly. The result is a set of comprehensive and "fair" evaluative questions, designed to effectively assess multimodal agricultural understanding.

## B.4    Final Stage: Human Verification

To guarantee the quality of the evaluation questions, we implemented a human verification process that validates faithfulness, certainty, quality, and MCQ feasibility. The data was distributed through an HTML file containing AGMMU questions and answers, original user questions, expert answers, and corresponding images. Each annotator was given a corresponding Excel file where the user just has to mark false (uncheck the box) for each condition not met per question. To further assist the annotator, we provided a few complex examples of questions that meet and do not meet the requirements, functioning as in-context examples. After collecting these data, only the completely unproblematic ones (all boxes remain checked) were kept.

**Faithfulness:**    *Do you think the question, ground-truth, and context extract faithful information from the original farmer question?* Our questions are directly based on the original questions and this step functions as a sanity check ensuring the quality of our dataset. The annotator needs to read through the question and the original conversations between the user and the expert.

**Certainty:** *Is the expert certain about the answer?* As our ground truth answers are extracted from the expert answers, we only want to include those that are very certain. A higher certainty from the expert means that it is more likely to be correct. We observe that the behaviors of the annotator are to read the responses from the expert and look for keywords like "may," "not sure," "you have to go to a lab for further inspection."

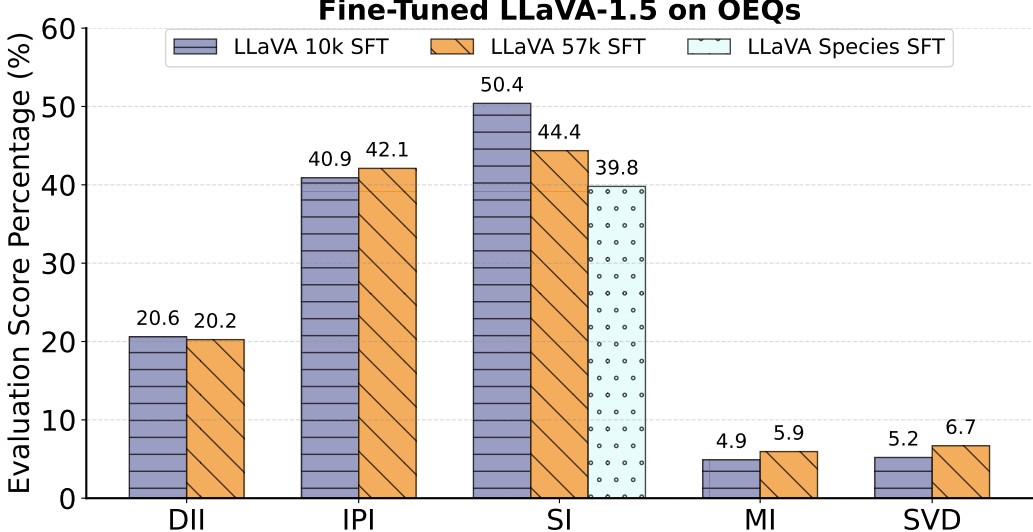

Figure J: Evaluation scores across five error categories for three fine-tuning setups using the AG-BASE dataset. Models are fine-tuned on: (1) **LLaVA 10k SFT**—a mix of AGBASE and 10k LLaVA-Instruct samples, (2) **LLaVA 57k SFT**—a 50-50 blend of AGBASE and LLaVA's original 57k SFT samples, and (3) **LLaVA Species SFT**—a specialized set focused on species identification with contextual augmentation.

**Quality:** *Are the images suitable for answering the questions?* (Images are not in low-resolution, blurs, pure blackness, etc.) *Are the image/symptom descriptions visible in the presented images?* As our benchmark attempts to evaluate the visual understanding of the models, our human verification removes the questions that do not depend on the images and those with broken images. For example, it is not fair for our question to ask about the fruit of the plant when the submitted photos only capture the leaves of the tree or the image is blurry.

**Feasibility:** *Are all of the wrong choices wrong?* The incorrect choices were generated with GPT-4o, so we need to check to ensure there are no multiple correct answers or an answer that overlaps with the correct answer and remove. For example, the wrong choice might be the common name of a species displayed by the scientific name in the ground truth.

## C More Evaluation, Implementation, and Design Choices

**LLM-as-judge.** To perform evaluation on few-word and multi-statement OEQ responses, we implemented the LLM-as-judge methodology using GPT-4.1. Our prompts for few-word responses (Figure E, F) and multi-statement responses (Figure G, H) contain several in-context examples based on the question category to guide the LLM to correctly categorize the answer as "correct," "incorrect," "partially correct," and "irrelevant."

**Number of MCQ options.** To determine the optimal number of answer choices for our MCQs, we conducted an ablation study comparing GPT-4o's accuracy when presented with four, five, and six options. For efficiency, we conducted this experiment on a subset of 821 questions, generating 5 wrong answers with GPT-4o. We randomly choose 3, and 4 wrong answers, for the four-choice and five-choice experiment, respectively, and take all choices for the six-choice experiment. While this limited subset may not capture the full variability of the dataset, it provides sufficient evidence to inform our design decisions. Due to the risk of process of elimination with MCQs, we believe that OEQs more accurately capture model performance.

The results, shown in Figure I, indicate that accuracy decreases as the number of answer choices increases. Specifically, we observed a 5-10% reduction in accuracy between the 4-option and 6-option configurations. This trend suggests that the model might rely on a process of elimination when selecting an answer, making it more challenging to identify the correct response as the number

of options increases. While the decrease in accuracy is not overly significant, we think it justifies our choice to use four options for MCQs.

**Implementation of AGBASE fine-tuning.** We fine-tune the LLaVA-v1.5-7B model using a LoRA-based setup. The training is performed with a learning rate of 2e-4, without weight decay, and a cosine learning rate schedule with a 3% warm-up ratio. We use a per-device batch size of 16 with gradient accumulation steps set to 2, resulting in an effective batch size of 64. The model is trained over 2 epochs using 2 NVIDIA A6000 GPUs.

Dataset preparation involved curating structured multi-turn conversations from a horticultural FAQ knowledge base, paired with user-uploaded images. From an initial pool of 367,331 QA-image pairs, we filtered out questions that had a species value in [*tree*, *bee*, *shrub*, *weed*, *wasp*, *plant*, *insect*, *grass*, *none*, *moth*, *beetle*, *snake*, *caterpillar*, *spider*, *ant*, *mushroom*, *fungus*], because we observe that questions with these common non-species species extractions often contain vague or uncertain examples. This gives us a high-quality dataset of 57,079 samples. Considering the influence of data mixture for training VLMs, we conduct three fine-tuning experiments. (1) The first experiment involves fine-tuning on a combination of our domain-specific dataset, AGBASE, and 10,000 samples from LLaVA's original instruction-tuning dataset, LLaVA-Instruct-150K. (2) The second experiment employs a 50-50 mixture of AGBASE by using 57,079 samples from LLaVA's original SFT set [25] (3) The third experiment focuses solely on species identification and consists of 18,109 QA pairs constructed by prepending the full original user queries to 33,777 generic identification samples, allowing us to test the effect of user context on classification accuracy.

In Figure J, we find that the LLaVA 10k SFT model achieves a slightly higher overall accuracy (0.25) compared to the LLaVA 57k SFT model (0.24), suggesting that a smaller, well-curated dataset mixed with domain-specific data may be more effective than a larger, more generic one for knowledge-intensive domain fine-tuning. Additionally, the LLaVA Species SFT model, which includes added user query context for species identification, performs worse than the other models in the species category , indicating that this additional context provides limited benefit for classification accuracy.

## D   More Dataset Visualization

In Figure K, we demonstrate more samples in AGMMU with questions and **multiple choice answers**.

In Figure L, we demonstrate more samples in AGMMU with **open-ended questions** and responses. We especially emphasize the long-form responses required from the model for symptom description and management instructions, normally containing multiple facts.

## E   Limitations and Future Work

While our work makes unique contributions to agricultural benchmark development and VLM evaluation through knowledge-intensive tasks, we acknowledge several limitations and identify promising directions for future research in this section.

**Advanced Utilization of Training Data.** Although our curated dataset, AGBASE, has proven significant effectiveness for fine-tuning VLMs [25] as shown in Section 4 and Figure 6, its potential extends beyond our current usage. As a comprehensive knowledge repository, the dataset presents opportunities for knowledge retrieval and augmented generation (RAG) approaches [5]. In particular, the development of vision-centric multimodal RAG systems remains an under-explored yet promising direction. This alternative could enable more effective knowledge extraction and utilization from our dataset, potentially improving model performance on agricultural understanding tasks. We leave the exploration of these advanced techniques for future work.

**Expanded Model Coverage and Evaluation Protocols.** While our current study encompasses several state-of-the-art and most commonly used VLMs for zero-shot evaluation and fine-tuning analysis, we acknowledge that they represent only a subset of available multimodal architectures and methodologies. To enhance the robustness and generalizability of our findings, we plan to incorporate a broader spectrum of VLMs. Additionally, we plan to conduct more extensive ablation studies and comparative analyses across different model scales and architectures. This comprehen-

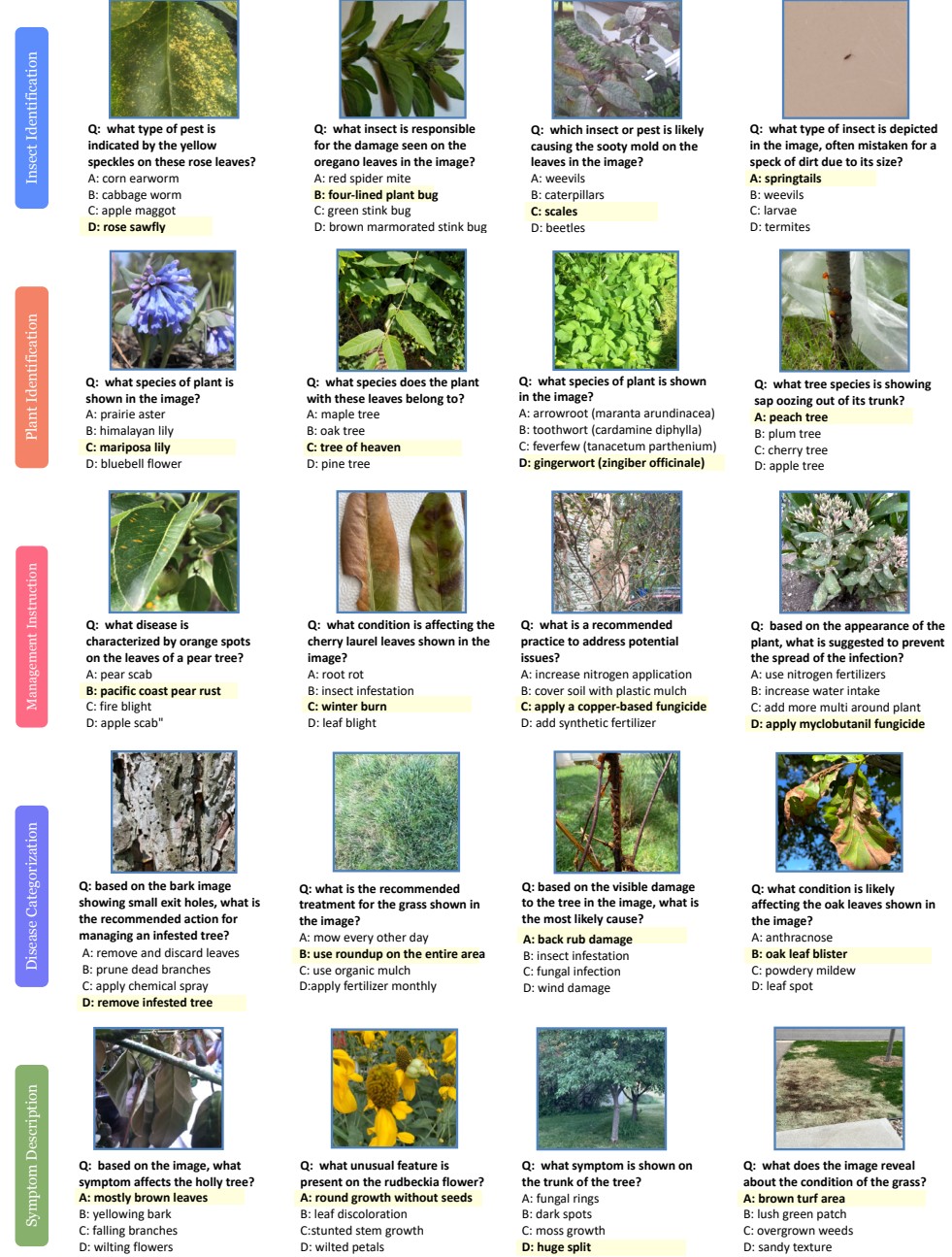

Figure K: Additional visualization of samples in AGMMU. Ground truth selections of each question are highlighted in yellow.

sive evaluation will provide deeper insights into the relative strengths and limitations of various approaches in agricultural understanding tasks.

# F  Societal Impact

We anticipate no direct negative societal impact of our work. Our dataset is ethically designed, respecting the privacy of Extension.org users by removing personal identifying information such as name, gender, username, and location. Additionally, we have verified to the best of our ability to ensure the removal of images that contain human faces. During dataset curation, we put in great

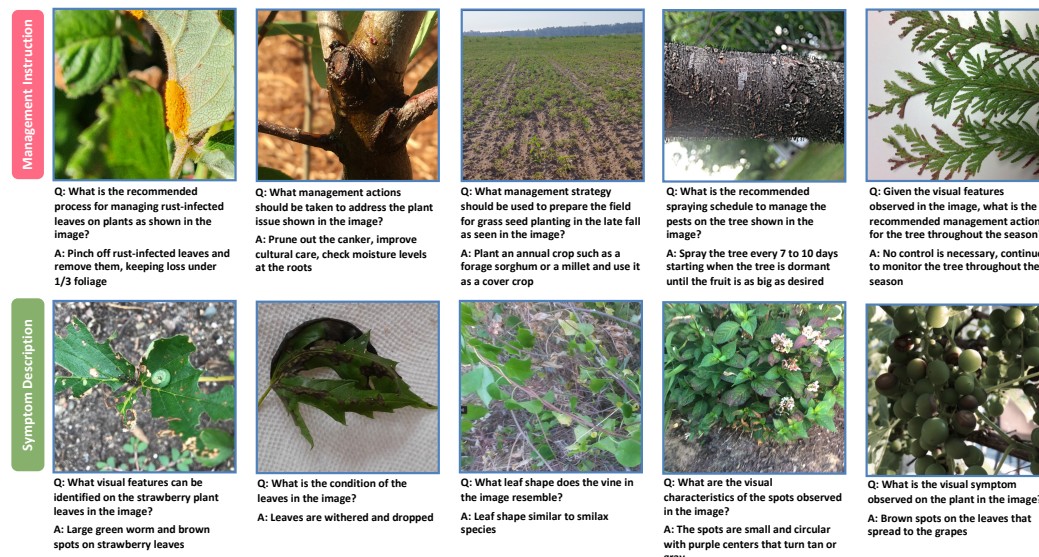

Figure L: Additional visualization of OEQ samples in AGMMU.

effort to eliminate bias by creating a dataset representative of the original Extension.org questions as well as a balanced dataset across all question types.

**Positive Impact:** We hope that the creation and release of this challenging vision-knowledge intensive dataset can support active research in this domain. Our comprehensive dataset is adapted from real-world conversations between users and experts, creating samples that are more representative of questions and images one may ask. This enables more accurate responses as demonstrated by our fine-tuning experiments. This dataset can be used to support the development of an agricultural vision language model that can provide users with instant assistance on various topics like insect/pest identification, disease categorization, and most importantly, management instructions. When properly used, these models have the potential to assist sustainability goals, prevent yield loss, and improve resource use.

