# Supplementary Materials

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

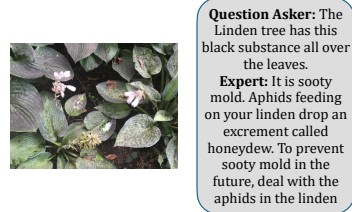

**Question Asker:** The Linden tree has this black substance all over the leaves.
**Expert:** It is sooty mold. Aphids feeding on your linden drop an excrement called honeydew. To prevent sooty mold in the future, deal with the aphids in the linden

**Step 2: Llama-405B Knowledge Extraction**

System Prompt: "*You are an assistant whose job it is to extract categories of information from an agriculture question. These categories are "disease/issue identification", "symptom description", "management instructions", and "miscellaneous facts". Make sure that "symptom descriptions" describe the visual symptoms that can be referenced in the hypothetical image. Make sure miscellaneous facts are independent of the exact situation and are standalone facts that don't depend on location or the time of year. If there is no information about any category, or if the asker/expert seems uncertain about their diagnosis, or simply directs the user to a diagnosis lab, tag it as "none". Format the response as a json. Here are some examples {examples}. Follow this format exactly. You will be given CONTENT header and you will ALWAYS include the word "TAGS:" as a header in your response so that your output format is always standardized, which is very important to the user.*"

Output:
TAGS:
{""*disease/issue identification*": "sooty mold", "*symptom description*": "black film/ substance all over the leaves, damage to leaves", "*management instructions*": "watch for and deal with aphids in the linden tree to prevent sooty mold in the future", "*miscellaneous facts*": [ "aphids feeding on linden trees drop an excrement called honeydew which is an ideal growing medium for sooty mold", "sooty mold can weaken the plant by reducing the photosynthetic layer" ] }"}

**Step 1: LLama-70B Agriculture Subdomain Categorization**

System Prompt: "*You are a helpful assistant tasked with categorizing farming-related questions by their entity and question type. Identify the primary LIVING entity that is being affected by the concern in the question, or 'none' if there is no primary living entity or it is unknown/can't be identified. This entity can be as broad as 'plant' or 'tree' but try to extract the most specific name of the entity mentioned. The question type categories to use are: 'disease', 'weeds/invasive plants management', 'insects/pests control', 'growing advice,' 'environmental stress','nutrient deficiency', 'generic identification', or 'other'. \n'insect control' is for any question that is related to insect issues. 'disease' is for any question about a disease or virus. 'growing advice' is for any question about how to grow or take care of a plant. 'environmental stress' is for any questions that pertain to problems caused by the environment such as heat. 'nutrient deficiency' is for problems that are related to nutrient deficiencies like fertilizers. 'generic identification' is for questions that are purely for entity identification, with nothing related to management or other issues. Only categorize in 'other' as a LAST RESORT. Here are some examples for each category: {examples} \nFollow this format exactly. you will be given CONTENT and you will ALWAYS include the header ENTITY: and CATEGORY: in your response so that your output format is always important to the user.*""

Output:
ENTITY: linden tree
CATEGORY: disease

**Step 3: GPT-4o MCQ Generation**

System Prompt: "*Your job is to convert information into challenging multiple choice question and answers. For each header, create one MCQs with 3 incorrect answers and 1 correct answer. make a VISION DEPENDENT question using the 'management instructions' header as the ground truth. You can include context your are given in the question, but try not to, especially make sure that the question doesn't contain info that gives away the correct answer. Make sure that the question is standalone and answerable given the image and question but still make the question succinct, with no more info than necessary. If this is not possible simply output 'none'. Here is what the format should look like {managamenet instructions: {'question': '<question here>', 'options': ['<option1>', '<option2>', '<option3>', '<option4>'], 'answer': '<answer>'}} make sure that the wrong choices have similar word length as the right answer. And though the wrong answers should be challenging, they should still be clearly incorrect so there is only one right answer. You will always be given the CONTENT header and you will always include QUESTIONS: as a header in your response so your response is standardized which is important to the user. CONTENT: context: symptom description – black film/substance all over the leaves, damage to leaves, species – linden tree,
'management instructions': 'watch for and deal with aphids in the linden tree to prevent sooty mold in the future'*"

Output:
QUESTION:
{"*question*": "what is one way to prevent the black substance observed on the leaves?", "*options*": [ "prune the linden tree weekly", "use fertilizer biannually", "increase water supply to roots", "apply aphid control measures" ], "*answer*": "apply aphid control measures"}

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

### Growing Advice

You are an assistant whose job it is to extract categories of information from an agriculture question. These categories are "succinct question", "succinct answer", "image description", "other management advice", and "miscellaneous facts". Make sure that "image descriptions" describe the visual symptoms that can be referenced in the hypothetical image. Make sure miscellaneous facts are independent of the exact situation and are standalone facts that don't depend on location or the time of year. If there is no information about any category, or if the asker/expert seems very uncertain about the information, tag it as "none". Format the response as a json. Here are some examples {examples}. Follow this format exactly. You will be given CONTENT header and you will ALWAYS include the word "TAGS:" as a header in your response so that your output format is always standardized, which is very important to the user.

### Insects/Pest Control

You are an assistant whose job it is to extract categories of information from an agriculture question. These categories are "insect/pest", "image description", "management instructions", and "miscellaneous facts". Make sure that "image descriptions" describe the visual qualities that can be referenced in the hypothetical image. Make sure miscellaneous facts are independent of the exact situation and are standalone facts that don't depend on location or the time of year. If there is no information about any category, or if the asker/expert seems very uncertain about the information, tag it as "none". Format the response as a json. Here are some examples {examples}. Follow this format exactly. You will be given CONTENT header and you will ALWAYS include the word "TAGS:" as a header in your response so that your output format is always standardized, which is very important to the user.'

### Weeds/Invasive Species Management

You are an assistant whose job it is to extract categories of information from an agriculture question. These categories are "image description", "management instructions", and "miscellaneous facts". Make sure that "symptom descriptions" describe the visual symptoms that can be referenced in the hypothetical image. Make sure miscellaneous facts are independent of the exact situation and are standalone facts that don't depend on location or the time of year. If there is no information about any category, or if the asker/expert seems very uncertain about the information, tag it as "none". Format the response as a json. Here are some examples {examples}. Follow this format exactly. You will be given CONTENT header and you will ALWAYS include the word "TAGS:" as a header in your response so that your output format is always standardized, which is very important to the user.' else: return None

### Disease/Environmental Stress/Nutrient Deficiency

You are an assistant whose job it is to extract categories of information from an agriculture question. These categories are "disease/issue identification", "symptom description", "management instructions", and "miscellaneous facts". Make sure that "symptom descriptions" describe the visual symptoms that can be referenced in the hypothetical image. Make sure miscellaneous facts are independent of the exact situation and are standalone facts that don't depend on location or the time of year. If there is no information about any category, or if the asker/expert seems uncertain about their diagnosis, or simply directs the user to a diagnosis lab, tag it as "none". Format the response as a json. Here are some examples {examples}. Follow this format exactly. You will be given CONTENT header and you will ALWAYS include the word "TAGS:" as a header in your response so that your output format is always standardized, which is very important to the user.