# OpenReview forum: "AgMMU: A Comprehensive Agricultural Multimodal Understanding Benchmark"
_NeurIPS.cc/2025/Datasets_and_Benchmarks_Track — NeurIPS 2025 Datasets and Benchmarks Track poster_

### Official Review · Reviewer_LPAt · 2025-06-13

**Rating:** 5
**Confidence:** 3

**Summary:**

This paper introduces AGMMU, a multimodal benchmark designed to evaluate vision-language models (VLMs) in the agricultural domain. The dataset is based on over 116,000 real-world conversations between users and USDA-certified agricultural experts. It includes 746 multiple-choice questions (MCQs), 746 open-ended questions (OEQs), and over 57,000 factual entries (AGBASE) for fine-tuning. The authors also benchmark several leading VLMs using this dataset.

**Additional Feedback:**

This work makes a valuable and timely contribution by creating the first real-world benchmark for agricultural VLMs. It offers high-quality data, diverse tasks, and solid experimental validation. Despite some limitations, this paper stands out in its novelty and impact.

**Dataset Code Accessibility:**

Yes

**Ethical Considerations:**

No, there are no or only very minor ethics concerns

**Final Justification:**

Thanks for the author's rebuttal. Most of my concerns have been addressed. I keep my rating.

**Limitations Weaknesses:**

1. While the paper presents a qualitative and categorical error analysis, the statistical rigor (e.g., confidence intervals or variance across multiple runs) is limited due to computational constraints.
2. The fine-tuning experiments are limited to a single open-source model (LLaVA-1.5). Additional results across more models or ablation studies (e.g., domain-specific subsets of AGBASE) would strengthen claims about the utility of the dataset for model improvement.
3. It remains unclear how well models fine-tuned on AGBASE generalize to unseen agricultural tasks or other related domains such as forestry or animal health, which may share similar multimodal characteristics.
4. While the inclusion of user-uploaded images adds realism, it also introduces noise and quality variance that could confound model performance. A deeper analysis of how image quality impacts performance would be beneficial.

**Strengths Contributions:**

1. All samples come from real user-expert conversations, ensuring high domain relevance and quality.
2. The dataset includes five key agricultural topics — pest identification, species identification, disease diagnosis, symptom description, and management advice.
3. Combines MCQs and OEQs to assess both perception and factual recall ability of models.
4. Code and data are publicly available for reproducibility and community use.

---

> ### Author Rebuttal · Authors · 2025-07-31
>
> We thank you for recognizing our work as valuable and timely, with high-quality data and diverse tasks for the crucial challenge of real-world agriculture. We are especially glad that our use of real-world user-expert conversations, the combination of open-ended and multiple-choice questions, and our publicly available data are well appreciated. We have conducted additional experiments and offer the following discussions in response to your feedback.
>
> ---
>
> 1. **Statistical Stability**
>
> Thank you for acknowledging the computational challenges involved in evaluating large models multiple times. While single-run evaluations are a common practice in VLM research due to cost, we agree that assessing statistical rigor is important.
>
> To address this, we conducted three full fine-tuning runs of our LLaVA-1.5-FT model (Sec. 4.5) and calculated the variance in performance. The results show very low variance, indicating that performance on AgMMU is stable.
>
> | Metric    | Run 1|Run 2 |Run 3 |Mean  | Std. Dev.    |
> | --------- | ---- | ---- | ---- | ---- | ---- |
> | OEQ Score |0.2466|0.2445|0.2408|0.2440|0.0024|
> | MCQ Score |0.6408|0.6421|0.6287|0.6372|0.0060|
>
> The low standard deviation, particularly for the more challenging OEQ set, demonstrates that our results are statistically significant and reliable.
>
> ---
>
> 2. **Additional Fine-tuning Experiments**
>
> To further demonstrate the utility of AgBase, we performed an additional fine-tuning experiment using Qwen-VL-7B. We adopted a similar fine-tuning setup used for LLaVA-1.5.
>
> As shown below, fine-tuning on AgBase provides a substantial performance boost to Qwen-VL on the OEQ set, nearly doubling its average score. This reinforces the value of AgBase for improving the agricultural knowledge of different open-source VLMs.
>
> | Category (OEQs)   | Qwen-VL | Qwen-VL-FT |
> |---------------------------|---------|--------------------|
> | Disease/Issue Identification | 10.2    | 23.5               |
> | Insect/Pest               | 25.2    | 49.0               |
> | Species                   | 36.1    | 61.6               |
> | Management Instructions   | 4.1     | 3.9                |
> | Symptom/Visual Description| 6.6     | 16.4               |
> | **Average**               | **16.4** | **32.9**           |
>
> ---
>
> 3. **Generalization**
>
> This is a critical point. While AgBase covers five core agricultural topics, assessing generalization is key. As a preliminary test, we evaluated our fine-tuned LLaVA model on a subset of the PlantDoc dataset, which it has never seen before. The task was open-ended disease identification via direct text matching.
>
> On 146 unseen examples from PlantDoc, the fine-tuned LLaVA model achieved an accuracy of 37.0%, a significant improvement over the base model's 8.9%.
>
> This result provides initial evidence that knowledge learned from AgBase can generalize to related, out-of-domain tasks. While more comprehensive cross-domain evaluation (e.g., on forestry or animal health) is an important direction for future work, these findings are promising.
>
> ---
>
> 4. **Image Quality**
>
> Thank you for highlighting the realism our user-uploaded images provide. We agree that analyzing the impact of image quality is an interesting research question. However, we also believe that for a real-world benchmark, it is essential to evaluate models on the noisy, variable-quality images they will encounter in practice. Our goal was to build a benchmark that reflects this reality.
>
> To ensure that poor image quality does not unfairly penalize models, we implemented a rigorous two-step quality control process:
>
> * **Expert-Driven Filtering**: During data collection, we automatically filtered out conversations where the expert expressed uncertainty or could not make a diagnosis, which often correlates with poor-quality images (see Fig. A, Step 2).
> * **Human Verification**: Our annotators were explicitly instructed to verify image quality, ensuring that images were not low-resolution, blurry, or otherwise indecipherable, and that the relevant symptoms were visible (see Supp. L157-L162).
>
> Through this process, we maintain the realism of user-submitted data while ensuring a high-quality, faithful evaluation. We invite the reviewer to browse the image previews on our Hugging Face dataset page, which demonstrate that the images are clear and informative.

---

### Official Review · Reviewer_U8ZK · 2025-06-26

**Rating:** 5
**Confidence:** 4

**Summary:**

This paper presents AGMMU, a real-world benchmark for evaluating VLMs in the knowledge-intensive domain of agriculture.

**Dataset Code Accessibility:**

Yes

**Dataset Code Comments:**

The data and annotations are available in the hugging face link. The descriptions are clear.

**Ethical Considerations:**

No, there are no or only very minor ethics concerns

**Final Justification:**

Based on comments from all reviewers, I will keep my rating

**Limitations Weaknesses:**

I cannot find any major weaknesses of this paper in terms of significance, writing quality, and experiments. But I do have a minor suggestion, that the format of some subsection titles can be more consistent. For example, the capitalization and the period at the end of the title: "Data source." vs. "AgMMU Overview.", "Step 3: QA Generation." vs. " Step4: Human verification").

**Strengths Contributions:**

1. The topic of benchmarking VLMs with knowledge-intensive domain of agriculture is meaningful and worth investigation.
2. The dataset is distilled from authentic dialogues between everyday growers and verified by experts, which is a significant contribution.
3. The dataset contains various tasks and topics, which can be interesting and useful.
4. The overall writing is clear and well-organized.

---

> ### Author Rebuttal · Authors · 2025-07-31
>
> Thank you for your positive and encouraging review of our paper. We are delighted that you found our work to be technically solid and recognized the significance of our contribution to benchmarking VLMs in the important domain of agriculture.
>
> We sincerely appreciate you pointing out the minor formatting inconsistencies in the subsection titles. This is an excellent observation, and you are correct that they should be standardized. We will carefully revise the manuscript and correct all formatting to ensure consistency in the final camera-ready version.
>
> Thank you again for your valuable feedback and your support for our work.

---

> ### Comment · Reviewer_U8ZK · 2025-08-06
>
> Based on comments from all reviewers, I think the authors have adequately addressed most of the concerns. So I will keep a positive rating.

---

### Official Review · Reviewer_QXmr · 2025-06-30

**Rating:** 5
**Confidence:** 4

**Summary:**

The paper introduces two multi-modal agricultural datasets: (1) AGMMU (746 samples) for open-ended agricultural knowledge evaluation, and (2) AGBase (57k samples) for fine-tuning vision-language models (VLMs) on agricultural knowledge. Both datasets are derived from real-world conversations and constructed through a detailed pipeline involving categorization, domain-specific knowledge extraction, QA generation, and human verification. The authors conduct zero-shot evaluations using 12 state-of-the-art VLMs and perform fine-tuning experiments with the LLaVA-1.5 model. The results highlight significant gaps in agricultural knowledge in current VLMs, while fine-tuning LLaVA-1.5 with AGBase leads to improved performance on AGMMU. The datasets are likely to be valuable and impactful for the agricultural AI research community.

**Additional Feedback:**

- It would be useful to provide relevant statistics for AGBase (e.g., topical categorization) and compare them to existing datasets.

**Dataset Code Accessibility:**

Yes

**Ethical Considerations:**

No, there are no or only very minor ethics concerns

**Final Justification:**

1) The authors have addressed my concerns regarding (a) the qualifications of human annotators and (b) the additional fine-tuning results and topical coverage of the dataset in their rebuttal. **[+0.5 rating]**

2) Regarding the use of the modified SimpleQA metric (originally not designed for partial credit)—the readers would benefit from an intuitive explanation of its behavior across a range of inputs. For example, does the modified metric reduce to the original SimpleQA metric when no *partially correct* answers are present in the judgments?
Additionally, why should such a metric be preferred over rubric-based or point-based alternatives, which may be easier to interpret in this setting? **[+0.5 rating]**

I have increased my score by a point considering the above recommendations.

**Limitations Weaknesses:**

- I am curious about the qualifications of the human annotators mentioned in Step 4 of Section 3.2.

- Line 272 states: "use the harmonic mean... Correct / (Total - Partial Correct), similar to SimpleQA." However, this differs from SimpleQA, which uses Correct / Attempted, i.e., (Correct + Incorrect). Could you clarify this discrepancy and explain how it might have impacted the results?

- It is unclear how fine-tuning on existing datasets such as CROP [1] and CDDM [2] would improve performance on AGMMU.

[1] Zhang, Hang, et al. "Empowering and assessing the utility of large language models in crop science." Advances in Neural Information Processing Systems 37 (2024): 52670-52722.

[2] Liu, Xiang, et al. "A Multimodal Benchmark Dataset and Model for Crop Disease Diagnosis." European Conference on Computer Vision. Cham: Springer Nature Switzerland, 2024.

**Strengths Contributions:**

- The paper is well-written and clearly motivates the design decisions behind the dataset curation and generation.

- It contributes a realistic multi-modal dataset in the agriculture domain, which is underrepresented and requires greater coverage.

- The authors provide both open-ended and multiple-choice question (MCQ) formats for evaluation.

- The paper includes a categorized error analysis that can help identify areas for improvement in current VLMs.

---

> ### Author Rebuttal · Authors · 2025-07-31
>
> Thank you for your thorough review and for recognizing the value of our work in providing a realistic, multi-format dataset for the underrepresented domain of agriculture. Your feedback is very helpful, and we are happy to provide the following clarifications to address your questions.
>
> ---
>
> 1. **Qualifications of Human Annotators**
>
> This is an important point for ensuring data quality. The knowledge in our benchmark is sourced directly from dialogues with authorized experts, so the primary role of our human annotators was not to be agricultural experts themselves, but to ensure the fidelity of the data pipeline. Specifically, their tasks were to verify:
> 1. That the automated knowledge extraction from the expert's answer was accurate.
> 2. That the generated question was coherent, unambiguous, and answerable from the provided image(s).
> 3. That the images were of sufficient quality for a VLM to analyze.
>
> To meet these requirements, we hired annotators with graduate-level research and engineering backgrounds in computer vision and vision-language models. This ensured they had the technical expertise to assess the quality and suitability of the multimodal samples for evaluating VLMs. Furthermore, all annotators completed a mandatory training session to standardize the verification process and ensure high inter-annotator agreement in argicultural tasks.
>
> ---
>
> 2. **Clarification on the Harmonic Mean Metric**
>
> Thank you for pointing out this discrepancy. You are correct that our metric intentionally differs from the standard SimpleQA formula. We introduced a "Partially Correct" category to handle the inherent nuances of agricultural knowledge, where answers are not always strictly binary. For example:
> * **Taxonomic Hierarchy**: In disease identification, guessing “fungus” when the correct answer is "black jelly fungus" is considered partially correct.
> * **Semantic Granularity**: In management instructions, suggesting "pesticide" when the correct answer is "spray insecticidal soap, carbaryl, or spinosad" is partially correct because pesticide includes the correct answer despite a specificity difference.
>
> Our evaluation metric, the harmonic mean, is designed to be more precise than a simple accuracy score. By excluding "Partially Correct" answers from the numerator entirely, we only reward models for answers that are fully correct. This prevents models from receiving credit for overly general or incomplete responses, pushing the evaluation towards rewarding precise, actionable knowledge. We will clarify this rationale in the revised manuscript.
>
> ---
>
> 3. **Fine-tuning on CROP and CDDM**
>
> Thank you for the insightful question of connecting different datasets together! It is meaningful for existing foundation models to benefit from more data sources.
>
> However, we would like to first clarify several key differences between CROP [1], CDDM [2], and our AgMMU:
> * In terms of knowledge coverage, our AgMMU aims at including a wider range of topics and knowledge: CROP concentrates on rice and corns, where our AgMMU covers more plant and insect species; CDDM focuses on crop diseases, while our AgMMU also features more open-ended management instructions and symptom description questions.
> * In terms of data source, our AgMMU are distilled from authentic, real-world dialogues between laypeople and credentialed experts, instead of documents (CROP) or GPT-generated QA (CDDM). This provides a unique and challenging data distribution that reflects the messy, practical problems faced in the field, a signal that is not present in the other datasets.
>
> Therefore, our AgMMU exhibits distinct distributions compared with CROP and CDDM. Such unique nature is also our decision of fine-tuning only with AgBase as our baseline in the paper.
>
> Despite such distinctions, we conducted an experiment fine-tuning LLaVA-1.5-7B on a combined dataset comprising 28.5k samples from CDDM [2] and 28.5k samples from our AgMMU training set.
>
> We followed the exact same fine-tuning and evaluation protocols as in our original study. The evaluation was performed on the AgMMU OEQ benchmark.
> | Category                | LLaVA-1.5<br>(Original) | Fine-Tuned<br>on AgMMU | Fine-Tuned<br>on AgMMU+CDDM |
> |-------------------------|-------------------------|--------------------------|-----------------------------|
> | Disease/Issue ID        | 4.9                     | 20.6                    | 16.9                        |
> | Insect/Pest             | 22.3                    | 40.9                    | 40.8                        |
> | Species Identification  | 30.6                    | 50.4                    | 43.4                        |
> | Management Instructions | 1.4                     | 4.9                     | 4.1                         |
> | Symptom/Visual Desc.    | 5.0                     | 5.2                     | 9.3                         |
>
> As seen in the table, fine-tuning with AgMMU alone consistently yields stronger results than the model fine-tuned on the AgMMU+CDDM mixture in most categories except symptom/visual description (SVD).
>
> ---
>
> 4. **Topical Coverage in AgBase**
>
> Thank you for this suggestion. Providing statistics for AgBase is important for contextualizing its contribution. We compare AgBase with several agricultural datasets that explicitly discussing the number of species/diseases or coverage:
>
> | Dataset       | Topological Coverage     |
> | ------------- | ------------------------ |
> | CROP [1]      | Primarily Rice and Corn  |
> | CDDM [2]      | 16 Crops and 60 Diseases |
> | AgriLLaVA [3] | 221 Pests/Diseases       |
> | AgBase (Ours) | 10164 Species, 5071 Pests, 4545 Diseases      |
>
> From the table, we conclude that AgBase offers the unique advantage of a wider coverage than the other datasets, which stems from its colelction from real-world user-expert conversations. We hope these clarifications have addressed your concerns and strengthened the case for our paper.
>
> [2] Liu et al. A Multimodal Benchmark Dataset and Model for Crop Disease Diagnosis. ECCV 2024.
> [3] Wang et al. Agri-LLaVA: Knowledge-Infused Large Multimodal Assistant on Agricultural Pests and Diseases.

---

> > ### Comment · Reviewer_QXmr · 2025-08-02
> >
> > I thank the author for their rebuttal addressing my concerns regarding (a) the qualifications of human annotators and (b) the additional fine-tuning results and topical coverage of the dataset.
> >
> > Regarding the use of the modified SimpleQA metric (originally not designed for partial credit)—the readers would benefit from an intuitive explanation of its behavior across a range of inputs. For example, does the modified metric reduce to the original SimpleQA metric when no *partially correct* answers are present in the judgments? Additionally, why should such a metric be preferred over rubric-based or point-based alternatives, which may be easier to interpret in this setting?

---

> > > ### Author Response · Authors · 2025-08-03
> > >
> > > Thank you for the follow-up question and for pushing us to clarify this important methodological detail. We are happy to provide a more intuitive explanation of our metric's behavior and our rationale for its selection.
> > >
> > > ---
> > >
> > > **1. Reduction to the Standard Metric**
> > >
> > > To your first point: Yes, our metric reduces to the standard Correct / Attempted formula when no "Partially Correct" answers are present.
> > >
> > > ---
> > >
> > > **2. Rationale for Preferring This Metric over Point-Based Alternatives**
> > >
> > > Our primary goal is to create a benchmark that is scalable, objective, and fairly evaluates nuanced knowledge in the complex domain of agriculture. We chose our metric because it provides a more forgiving and stable evaluation than strict accuracy, while being more scalable than rubric-based scoring.
> > >
> > > * **Scalability and Objectivity**: First, compared to rubric-based or granular point-based scoring, our three-category system (Correct, Incorrect, Partially Correct) is more objective and allows for faster, more consistent annotation, which is crucial for a public benchmark.
> > > * **Fair Evaluation of Nuanced Knowledge**: The key intuition of our metric, Correct / (Total - Partial Correct), is to not penalize a model for being partially correct. In a knowledge-intensive domain, a response that is directionally correct (e.g., identifying a "fungus" when it's a specific "black jelly fungus") is far more desirable than a completely incorrect answer. Our metric judges the model only on the questions where a binary correct/incorrect judgment is clearer.
> > >
> > > This approach is preferable to a point-based system (e.g., 1 for correct, 0.5 for partial) because it is simpler to interpret: the score represents the model's accuracy on the questions for which a definitive answer was provided.
> > >
> > > We believe this metric provides a fair, stable, and scalable evaluation for the nuanced challenges presented in AgMMU. We are happy to add results based on other alternative metrics in our final revision for completion. We will also add this detailed explanation and example to the manuscript to ensure the rationale is clear to all readers.

---

> > > > ### Comment · Reviewer_QXmr · 2025-08-05
> > > > **Follow-up**
> > > >
> > > > Thank you for your response, for reiterating your objectives, and for considering alternative metrics.  There may be a misunderstanding regarding the formulation of the reported metric in the paper. Could you please confirm the values for both the metric used in your work and the original SimpleQA metric (as defined in Section 2.4 of the SimpleQA [1] paper), using the following example input?
> > > >
> > > > `correct_answers = 10, incorrect_answers = 5, partially_correct = 0, not_attempted = 20`
> > > >
> > > > [1]  Wei, Jason, et al. "Measuring short-form factuality in large language models." arXiv preprint arXiv:2411.04368 (2024).

---

> > > > > ### Author Response · Authors · 2025-08-06
> > > > >
> > > > > Thank you for your additional clarifications and follow-up with such a concrete example! We are glad to use this opportunity to fully address your questions.
> > > > >
> > > > > ---
> > > > >
> > > > > Given your examples, we realized that you are asking the questions regarding two aspects: (1) why we introduced the "partially correct" category, and (2) why we are not considering the "not attempted" category. We hope the clarifications in the previous rebuttal have explained the reason for introducing "partially correct." Therefore, we will mainly clarify why AgMMU does not specifically consider **not attempted**.
> > > > > 1. **Agricultural Use Case**. Agricultural questions from farmers emphasize instancy, so the behavior of "not attempted" should not be encouraged. Instead, the models are encouraged to make reasonable speculations (*i.e.*, partially correct) to provide guidance for farmers. In our metric computation, we treat **"not attempted" as "incorrect" because it does not benefit the farmers.**
> > > > > 2. **Observation on Models**. We closely observed the answers of open-ended questions from vision-language models, and witnessed very few cases of models refusing to answer on AgMMU: according to our analysis, the ratio is 0\% for GPTo4 and 1.25\% for Gemini-Pro. Combined with our treating "not attempted" as "incorrect" mentioned above, the low ratio of "not attempted" suggests that our design choice does not influence the fair comparison in our agricultural scenarios.
> > > > >
> > > > > To conclude, we exclude the complication of introducing "not attempted" as it is both impractical for agricultural applications and redundant for existing VLMs in agricultural questions.
> > > > >
> > > > > ---
> > > > >
> > > > > Coming back to your example, we use the following notations for the simplicity of equations:
> > > > >
> > > > > * Correct: C
> > > > > * Partially Correct: P
> > > > > * Incorrect: I
> > > > > * Not Attempt: N
> > > > >
> > > > > Then, SimpleQA's metric calculation in your example is that:
> > > > >
> > > > > $\frac{2} {(C + I + N) / C + (C + I) / C } = \frac{2C}{2C + 2I + N} =  \frac{20}{20 + 10 + 20} = 0.4$
> > > > >
> > > > > The score in AgMMU's way is to treat "not attempted" as "incorrect." With partially correcting being zero, the harmonic mean is equivalent to the accuracy:
> > > > >
> > > > > $\frac{C}{C + I +N + P} = \frac{10}{10 + 5 + 20} = 0.28$
> > > > >
> > > > > ---
> > > > >
> > > > > To make it clearer, let's consider the distribution closer to our observation in AgMMU: 10 Correct, 5 Incorrect, 0 Not Attempted, and 10 Partially Correct.
> > > > >
> > > > > SimpleQA's score is to consider "partially correct" as "incorrect," so the score is
> > > > >
> > > > > $\frac{2} { (C + I + N + P) / C + (C + I + P) / C} = \frac{2C}{2C + 2I + N + 2P} = \frac{20}{20 + 10 + 0 + 20} = 0.4$
> > > > >
> > > > > While our AgMMU's score is to penalize "partially correct" less than "incorrect" and treat "not attempted" as "incorrect," which is zero, so the score is
> > > > >
> > > > > $\frac{2}{(C + I + P) / C + (C + I) / C} = \frac{2C}{2C + 2I + P} =  \frac{20}{20 + 10 + 10} = 0.5$
> > > > >
> > > > > Therefore, we hope the above clarifications more thoroughly explain the difference between our focus and SimpleQA and justify our concentration on agricultural use cases.

---

> > > > > > ### Comment · Reviewer_QXmr · 2025-08-08
> > > > > >
> > > > > > I appreciate your clarification regarding the agricultural use case and the rationale for the derived metric. Some readers may view the ability of (V)LLMs to *abstain* from answering unknown questions as a desirable feature to reduce hallucinations. Explicitly stating these underlying assumptions and evaluation criteria in the revised version would help avoid ambiguities and ensure fair, consistent comparisons in future work.

---

> > > > > > > ### Author Response · Authors · 2025-08-08
> > > > > > >
> > > > > > > Thank you for your positive rating, continued feedback, and suggestions for our paper! This point of clarifying the underlying assumption and nuances of agricultural scenarios is very reasonable. We will integrate this discussion thoroughly into the part discussing evaluation metrics in the main paper.

---

### Official Review · Reviewer_3h1R · 2025-07-02

**Rating:** 5
**Confidence:** 4

**Summary:**

In this paper, the authors have created an evaluation set/benchmark (AgMMU) and development set (AgBase) built from hundreds of thousands of real conversations between users and agricultural domain experts. AgMMU includes question-answer pairs, for multiple choice and open-ended questions, while AgBase consists of 57K+ multimodal facts about different agricultural topics. 12 VLMs were evaluated as a part of this benchmark. The authors demonstrate that simple finetuning of the open source models with AgBase yields performance improvements, suggesting that there is room for further improvement with more investment in finetuning.

**Additional Feedback:**

Questions/comments
* Line 239: The authors mention that “our benchmark exhibits the necessity of multi-image understanding challenge”,  but my understanding is that there is only a single image associated with each QA pair (based on Figures K and L in the supplemental material)?
* I’m curious if the answers in the original conversational corpus would ever be expected to change? For example, if a new fertilizer or pesticide was introduced, the “management instructions” might change to account for this, or if a new disease was detected in crops. If so, how would you propose augmenting the data to account for this?
* The authors may consider putting the limitations discussion in the main paper, not in the supplemental material.

**Dataset Code Accessibility:**

Yes

**Dataset Code Comments:**

The benchmark and data are available at the provided URLs. It comes with code for running inference and evaluation. There are details in the supplemental material regarding prompt templates used in information extraction, which make this work more readily reproduced.

**Ethical Comments:**

The authors have made an effort to remove personally identifiable information, as well as faces, from the data. This processing step involved human verification, which gives the reader more confidence that the data was processed sufficiently.

**Ethical Considerations:**

No, there are no or only very minor ethics concerns

**Final Justification:**

Thank you to the authors for the rebuttal. I believe the responses have adequately addressed the noted weaknesses I had observed with the paper. I've also read the authors' thoughtful responses to the other reviewers' comments. I stand by my recommendation of this paper for acceptance.

**Limitations Weaknesses:**

* The proprietary models evaluated are (already) somewhat outdated, and I am curious how well the more capable models perform (Claude 3.5+, including Sonnet; Gemini 2.0/2.5, GPT-4o), especially given how challenging the benchmark is, and how use of CoT appears to boost performance. The authors have acknowledged model coverage as a limitation.
* It doesn’t appear like there was any human verification used with GPT-4.1, for its use as a judge for grading OEQ answers. I’m curious if there are instances where the model graded incorrectly. Since the same model is consistently used for all model responses (with different prompts for different categories, or types of response), it may not be as much of an issue.

**Strengths Contributions:**

* This area (agriculture) is one where domain expertise is necessary, so we need to evaluate both visual understanding and knowledge understanding in this context. (Notably, the benchmark is quite challenging and It’s an interesting, impactful application, and the methodology can be extended to other domains where domain expertise is similarly required.
* Based on the related work cited in the paper, this work appears to address a critical gap in current benchmarks for this domain.
* The data used for the eval benchmark and development dataset is drawn from a sizeable set of real world conversations and images. The answers to the questions are provided by domain experts, and the images come in a variety of qualities and resolutions.
* The inclusion of the development corpus, along with the evaluation data, helps encourage further research and investigation in this area..
* All QA pairs are manually verified by humans to ensure high quality evaluation data.

---

> ### Author Rebuttal · Authors · 2025-07-31
>
> Thank you for your positive and insightful feedback. We are encouraged that you recognize AgMMU's value in addressing a critical gap in a knowledge-intensive domain. We appreciate your constructive questions and will incorporate your suggestion to move the limitations section into the main paper in our final revision.
>
> ---
> 1. **Performance of Newer VLMs**
>
> Thank you for this suggestion. We agree that benchmarking against the latest models is crucial. We have now evaluated several state-of-the-art models on our challenging open-ended questions (OEQs) and will add these results to the paper.
>
> | Model          | Disease | Insect/Pest | Species | Management | Symptom | Ave  |
> | -------------- | ------- | ----------- | ------- | ---------- | ------- | ---- |
> | GPT-4o         | 43.8    | 49.6        | 58.2    | 15.2       | 5.7     | 34.5 |
> | Claude 3.5     | 44.4    | 37.6        | 48.4    | 13.0       | 4.5     | 29.6 |
> | Gemini 2.5 Pro | 50.0    | 61.1        | 77.1    | 14.0       | 7.1     | 41.9    |
>
> These results lead to two key observations:
> * **Performance Gains**: As expected, newer models like Gemini 2.5 Pro show improved performance, setting a new state-of-the-art on our benchmark.
> * **Persistent Challenges**: Despite these advances, the scores for management instructions and symptom description remain low across all models. This highlights the benchmark's difficulty and underscores the continued need for domain-specific fine-tuning to master nuanced agricultural knowledge.
>
> ---
>
> 2. **Reliability of the GPT-4.1 Judge**
>
> We agree that the reliability of our LLM judge is paramount. We ensured its robustness through a two-fold process: rigorous prompt engineering during development and a new verification study for this rebuttal.
>
> * **Empirical Human Agreement**: We randomly selected 50 responses generated by our fine-tuned LLaVA model (10 from each category) and had a human expert manually grade them. As shown below, the judgments from GPT-4.1 show a very high alignment with human evaluation.
>
> | Category                   | Correct / Total |
> |---------------------------|-----------------|
> | Disease/Issue Identification | 9 / 10        |
> | Insect/Pest               | 9 / 10          |
> | Species                   | 10 / 10         |
> | Management Instructions   | 10 / 10         |
> | Symptom/Visual Description| 9 / 10          |
>
> * **Nature of Disagreements**: The few mismatches were all borderline cases where GPT-4.1 assigned a "partially correct" label to an answer a human grader deemed "incorrect." For instance, for a ground truth of "leaf-spotting fungi or bacteria," the judge rated the prediction "bacterial wilt disease" as partially correct. Given the shared keyword ("bacteria") and overlapping symptoms, this judgment is defensible and not a clear failure of reasoning.
>
> * **Robust Evaluation Metric**: Crucially, our scoring metric is designed to be robust to such borderline cases. As detailed in L271-L272, credit is only given for fully correct answers, so these "partially correct" labels do not inflate the final performance scores.
>
> Together, the high empirical agreement with human experts and the conservative design of our evaluation metric confirm that GPT-4.1 serves as a reliable and consistent grader for AgMMU.
>
> ---
>
> 3. **Multiple Images**
>
> Thanks for your close observation! The figures are mostly for display purpose, and we specially show one image per question to illustrate more samples in the paper. However, the AgMMU benchmark explicitly includes questions that require reasoning over multiple images.
>
> These multi-image samples are present in the dataset, which is publicly available on Hugging Face. For example, in the file `agmmu_e_filtered_hf1.json`, you will find entries like the one below, which contains two source images for a single question:
>
> ```
>         "faq-id": 601861,
>         "qtype": "management instructions",
>         "images": [
>             "./images/601861/601861_1.png",
>             "./images/601861/601861_2.png"
>         ],
>         "question_background": "...",
>         "question": "What management practice should be employed to reduce the chance of anthracnose ...",
>         "options": [
>             "...
>         ],
>         "answer": "clean up the fallen leaves around the shrub"
> ```
> This design choice directly addresses the need for multi-image understanding, a common requirement in real-world agricultural diagnostics.
>
> 4. **Benchmark and Knowledge Base Maintenance**
>
> This is an excellent point. Maintaining a current and relevant knowledge base is critical for real-world applications. We have a clear roadmap for keeping AgMMU up-to-date.
>
> * **Continuous Data Pipeline**: Our partnership with the AskExtension program through the AIFARMS institute is ongoing. This provides us with a continuous stream of new, expert-verified conversations, which we will use to periodically refresh and expand the benchmark.
>
> * **Temporal Versioning**: Every question in our dataset includes a `question_background` field containing metadata, including the date of the inquiry. This temporal information is a key feature, enabling the benchmark to track the evolution of agricultural knowledge (e.g., new best practices, emerging diseases). It also opens up future research avenues for evaluating a model's ability to adapt its knowledge over time.
>
> We believe this strategy will ensure AgMMU remains a valuable and timely resource for the research community.

---

> > ### Comment · Reviewer_3h1R · 2025-08-05
> >
> > Thank you to the authors for the rebuttal. I believe the responses have adequately addressed the noted weaknesses I had observed with the paper. I've also read the authors' thoughtful responses to the other reviewers' comments. I stand by my recommendation of this paper for acceptance.

---

### Comment · Area_Chair_4yZK · 2025-08-01
**Responses to author rebuttal?**

Reviewers, can you please take a look at the author's rebuttal and respond as soon as possible.

---

### Note · Authors · 2025-08-13

We sincerely thank the Senior Area Chairs, Area Chairs, and all reviewers for their valuable time and constructive feedback throughout this process. We are greatly encouraged by the positive reception of our work from all reviewers.

Following our detailed rebuttal and the subsequent discussions, we are pleased that all reviewers have maintained their positive ratings. They have explicitly stated that our responses and new experimental results have successfully addressed their primary concerns.

The key additions we made in response to the discussion include:

* **Benchmarking Newer SOTA Models**: We evaluated Gemini 2.5 Pro, GPT-4o, and Claude 3.5. These results set a new performance bar while simultaneously confirming AgMMU’s persistent challenges, particularly in knowledge-intensive categories.

* **Validating Our Methodology**: We conducted a human verification study on our LLM judge to confirm its reliability, ran multiple trials to demonstrate the statistical stability of our fine-tuning results, and provided a detailed rationale for our evaluation metric.

* **Demonstrating AgBase’s Utility**: We expanded our experiments to further show AgBase's value by fine-tuning an additional model (Qwen-VL) and testing generalization on the unseen PlantDoc dataset, both of which yielded significant performance gains.

This collaborative process has substantially strengthened our paper. We will incorporate all of these discussions, clarifications, and new findings into the final camera-ready version to reflect the valuable feedback received. Thank you again for your engagement and support.

---

### Decision · Program_Chairs · 2025-09-18

**Decision:**

Accept (poster)

**Comment:**

All reviewers agree that this benchmark represents a valuable contribution to the community and recommend acceptance, citing that the benchmark appears to "address a critical gap in current benchmarks for this domain," provides a "realistic multi-modal dataset in the agriculture domain," and appreciated that the benchmark included multiple evaluation settings (open-ended vs multiple choice) and was built with domain experts.